# Structural and functional analysis of the promiscuous AcrB and AdeB efflux pumps suggests different drug binding mechanisms

Alina Ornik-Cha[1,6], Julia Wilhelm[1,6], Jessica Kobylka[1], Hanno Sjuts[1,5], Attilio V. Vargiu [2], Giuliano Malloci [2], Julian Reitz [3,4], Anja Seybert[3,4], Achilleas S. Frangakis [3,4✉] & Klaas M. Pos [1✉]

Upon antibiotic stress Gram-negative pathogens deploy resistance-nodulation-cell division-type tripartite efflux pumps. These include a $H^+$/drug antiporter module that recognizes structurally diverse substances, including antibiotics. Here, we show the 3.5 Å structure of subunit AdeB from the *Acinetobacter baumannii* AdeABC efflux pump solved by single-particle cryo-electron microscopy. The AdeB trimer adopts mainly a resting state with all protomers in a conformation devoid of transport channels or antibiotic binding sites. However, 10% of the protomers adopt a state where three transport channels lead to the closed substrate (deep) binding pocket. A comparison between drug binding of AdeB and *Escherichia coli* AcrB is made via activity analysis of 20 AdeB variants, selected on basis of side chain interactions with antibiotics observed in the AcrB periplasmic domain X-ray co-structures with fusidic acid (2.3 Å), doxycycline (2.1 Å) and levofloxacin (2.7 Å). AdeABC, compared to AcrAB-TolC, confers higher resistance to *E. coli* towards polyaromatic compounds and lower resistance towards antibiotic compounds.

[1] Institute of Biochemistry, Goethe-University Frankfurt, Max-von-Laue-Straße 9, 60438 Frankfurt am Main, Germany. [2] Department of Physics, University of Cagliari, 09042 Monserrato (CA), Italy. [3] Buchmann Institute for Molecular Life Sciences, Goethe-University Frankfurt, Max-von-Laue-Straße 15, 60438 Frankfurt am Main, Germany. [4] Institute of Biophysics, Goethe-University Frankfurt, Max-von-Laue-Straße 1, 60438 Frankfurt am Main, Germany. [5] Present address: Biologics Research, Sanofi-Aventis Deutschland GmbH, Frankfurt, Germany. [6] These authors contributed equally: Alina Ornik-Cha, Julia Wilhelm. ✉email: achilleas.frangakis@biophysik.org; pos@em.uni-frankfurt.de

The Gram-negative opportunistic pathogen *Acinetobacter baumannii* exhibits a high level of multidrug resistance (MDR) to drugs including carbapenems and the last-resort antibiotics tigecycline and colistin[1]. This feature, paired with its persistence in hospital settings, contributes to frequent nosocomial outbreaks of *A. baumannii* infection[2]. Following the increasing emergence of strains that are non-susceptible to all clinically used antibiotics, the WHO has ranked carbapenem-resistant *A. baumannii* first place in its global priority pathogen list[3].

The superfamily of resistance-nodulation-cell division (RND) efflux pumps plays a key role in intrinsic MDR in Gram-negative bacteria. These tripartite complexes comprise an RND transporter in the inner membrane that acts as a secondary active H$^+$/drug antiporter extruding a vast spectrum of structurally unrelated drugs through a periplasmic membrane fusion protein channel connected to an outer membrane channel factor across the outer membrane[4]. RND transporters of the Hydrophobe/Amphiphile Efflux-1 (HAE-1) family are composed of 12 transmembrane (TM) helices, which form the TM domain (TMD). Two loops emerging between TM1 and TM2 and between TM7 and TM8 form a large periplasmic region that is divided into the inner membrane-proximal porter domain and the distal funnel domain. The porter domain is built up by the four subdomains PN1, PN2, PC1 and PC2, while the funnel domain consists of the DN and DC subdomains[5].

The best characterized RND transporter is the HAE1 family member AcrB from *Escherichia coli*. The first solved structure of AcrB was a symmetric homotrimer comprising three protomers in the so-called loose (L) conformation (thus denoted as LLL)[5]. Later efforts yielded high-resolution crystallographic structures of asymmetric AcrB trimers adopting loose (L), tight (T) and open (O) conformations (LTO) in the apo state[6,7] and with bound substrate molecules[8–10]. Substrate binding was observed only for the L and T protomers, at the access pocket (AP) and deep-binding pocket (DBP), respectively. These binding pockets are connected to the periplasm via four channels (CH1–CH4)[6,7,11,12]. In the O conformation, the AP, DBP and all channels are closed, although there is an exit channel from the (closed) DBP site leading to the funnel domain[6,7]. Structures of the tripartite RND complexes AcrAB-TolC from *E. coli* and MexAB-OprM from *Pseudomonas aeruginosa* have also been determined using single-particle cryo-electron microscopy (cryo-EM)[13–15].

The current hypothesis is that any of the protomers within the AcrB trimer can adopt any of the three conformations L, T, and O[16]. During drug efflux, the protomers cycle through these states in a concerted and consecutive manner, so that drugs are sequestered from the periplasm (and the outer leaflet of the inner membrane), followed by binding at the AP and/or DBP. In the T conformation, protons can enter the TMD via water channels and protonate the charged residues D407 and/or D408, thus changing the electrostatics inside the TMD. This leads to a conformational change of the TMD and energy transduction towards the porter domain, which in turn leads to the closure of the AP, DBP, and CH1–CH4 channels and the opening of the exit channel. The efflux activity of AcrB is readily inhibited by molecules binding in an area of the DBP known as the hydrophobic trap[17]. Recently, we constructed and crystallized a soluble fusion of the two periplasmic AcrB loops (AcrBper) together with pyranopyridine-based inhibitors (e.g., MBX3132) and rhodamine 6G, which helped us expand our understanding of substrate and inhibitor recognition[18].

The three *A. baumannii* RND transporter complexes AdeABC, AdeFGH and AdeIJK were identified and characterized mainly in their natural host[19–21] but also heterologously in *E. coli*[22]. The constitutively expressed transporter complex AdeIJK is responsible primarily for intrinsic drug resistance in *A. baumannii*, with

overexpression showing cytotoxic effects[21]. AdeABC and AdeFGH are strongly regulated and are involved in acquired drug resistance[20,23]. Single point mutations in the regulators of both pump complex genes are enough to induce their strong over-expression in strains exposed to low concentrations of drugs[20,23,24]. AdeABC was found to be upregulated in most clinical MDR strains, whereas AdeFGH overexpression is less common[20,25]. Therefore, we focused our efforts on obtaining structural information for AdeB of *A. baumannii*. The broad substrate spectrum of AdeB, like that of *E. coli* AcrB[26], comprises β-lactams such as carbapenems and cephalosporins, fluoroquinolones, tetracyclines (including tigecycline), chloramphenicol, macrolides, trimethoprim, ethidium, rifampicin and novobiocin[19,22,27]. Unlike AcrB, however, AdeB was reported to confer resistance to aminoglycoside antibiotics in *A. baumannii* BM4454[19] and BM4689[24]. Others reported in both *A. baumannii* ATTC17978[27] and by heterologous expression in *E. coli*[22] that the resistance against aminoglycosides is not clearly attributable to the expression of the AdeABC efflux pump alone, and it was suggested that AdeABC overexpression observed in clinical strains appears essential, but not the sole factor for the increased resistance against aminoglycosides[27]. In this study, we aim to elucidate the previously uncharacterized determinants of substrate polyspecificity in AdeB. For structural elucidation of this RND transporter, we reconstituted AdeB in Salipro Nanodiscs sustaining a native-like lipid environment[28].

Here, we show the single-particle cryo-EM structure of trimeric AdeB in the OOO conformation and in the L*OO conformation. Molecular docking studies followed by free-energy calculations indicate binding of ethidium and rhodamine 6G to the L* protomer, suggesting a role of this conformational state in initial drug uptake in the overall catalytic drug transport mechanism. In addition, we analyze the role of key residues in the binding of nine different substrates, based in part on three crystal co-structures of AcrBper in complex with the known AcrB and AdeB substrates levofloxacin, doxycycline and fusidic acid. These three high-resolution structures show previously unknown binding modes and add to our understanding of the substrate promiscuity of RND transporters.

## Results

**Structural investigation of *A. baumannii* AdeB shows two distinct conformations**. AdeB from *A. baumannii* was produced in *E. coli* C43(DE3) Δ*acrAB* and purified by immobilized metal affinity chromatography (IMAC) followed by size exclusion chromatography (SEC). To determine the structure of *A. baumannii* AdeB in a native-like lipid environment, the protein was reconstituted into Salipro Nanodiscs (Supplementary Fig. 1). After confirming the integrity and trimeric state of the protein by native polyacrylamide gel-electrophoresis and negative stain electron microscopy (EM), samples were vitrified on holey carbon grids and analyzed by cryo-EM (Supplementary Fig. 2, statistics in Supplementary Table 1). A C3-symmetric density map could be solved up to a resolution of 3.5 Å (Fig. 1 and Supplementary Figs. 3 and 4). Our resulting structural model was compared to published structures of the RND transporters AdeB[29,30], AcrB[10], MexB[31], MtrD[32] and CusA[33] (Supplementary Table 2) and found to adopt the previously reported OOO conformation[29] (Figs. 1A, C and 2A, D). However, not all particles adopted this trimeric arrangement. After classification of all protomers, we found that approximately 10% of the protomers adopt an intermediate state, i. e. 30% of particles adopt a trimeric arrangement of two O protomers together with a previously uncharacterized conformation. Based on the comparison with the AcrB and AdeB structures (Supplementary Table 2), this protomer structure corresponds to neither L nor T conformations, hereafter referred

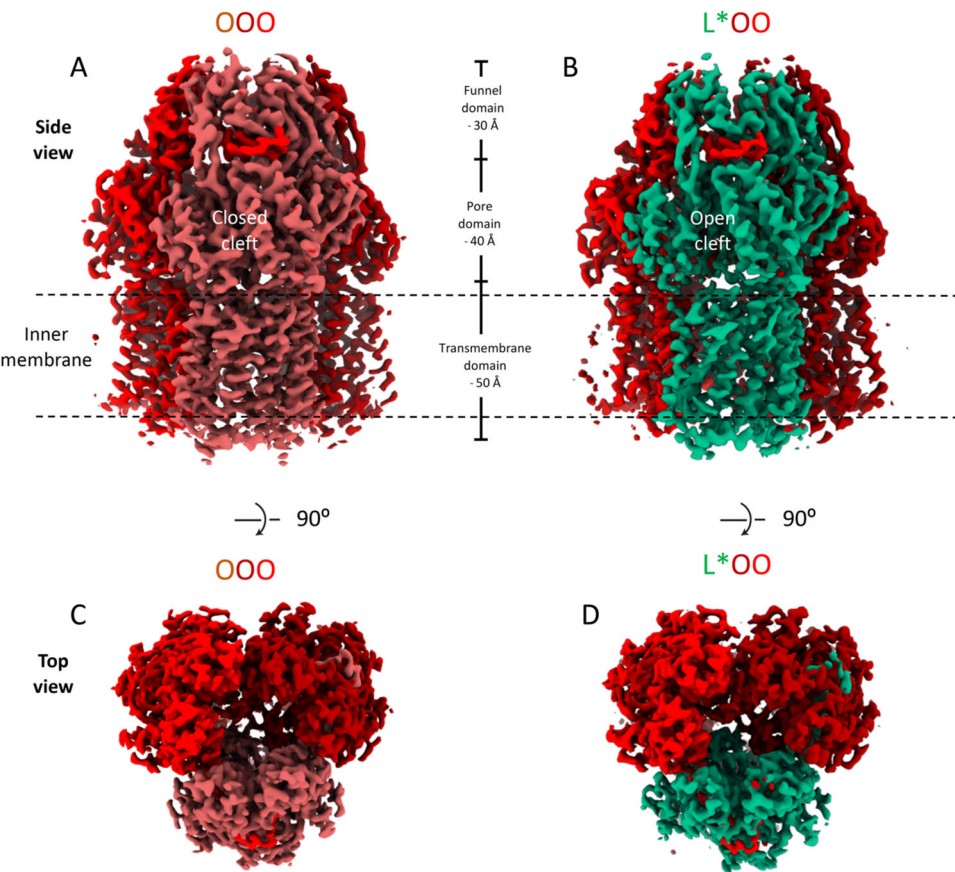

**Fig. 1 Cryo-EM density maps of AdeB. A** Side view of AdeB density in the OOO conformation at an overall resolution of 3.54 Å. **B** Side view of AdeB density in the L*OO conformation at 3.84 Å resolution. **C** Top view of AdeB density in the OOO conformation. **D** Top view of AdeB density in the L*OO conformation. The densities for the protomers are shown from light to dark red for the O conformation and in green for the L* conformation. The approximate boundary of the inner membrane embedded part of AdeB is indicated by dashed lines. The dimensions of the densities corresponding to the transmembrane, porter and funnel domains are indicated in (A) and (B). The densities are displayed at cutoff levels 0.0356 (OOO) and 0.0343 (L*OO) in ChimeraX (https://www.rbvi.ucsf.edu/chimerax).

to as L*, thus constituting the AdeB trimer L*OO (Figs. 1B and 2B, C and Supplementary Figs. 5 and 6). We solved the structure of AdeB in this newly described conformation with an overall resolution of 3.95 Å for the L* protomer and 3.84 Å for the L*OO trimer (Figs. 1 and 2, statistics in Supplementary Table 3 and Supplementary Figs. 5 and 6). The overall architecture of the L* protomer most closely resembles the T conformation of AcrB (overall RMSD ($C_\alpha$'s): 0.68 Å with the T conformation of AcrB, compared with 0.89 Å with the L conformation of AcrB)(Fig. 2G, H). Interestingly, the comparison of the AdeB L* protomer with the AdeB T protomer structures in the access/binding/extrusion (LTO, PDB: 7KGI) and the binding/extrusion/extrusion (TOO, PDB: 7KGG) conformations[30], indicated the lowest overall RMSD's between AdeB L* and other AdeB protomers of 1.81 Å and 1.55 Å, respectively (Supplementary Table 2 and Supplementary Fig. 6).

However, a unique arrangement of the PN2 and PC1 subdomains could be observed in the L* conformation. Whereas the structure of the PC1 subdomain, including the switch loop, is more similar to the L conformation of AcrB (local PC1 RMSD ($C_\alpha$'s): 1.32 Å with the T conformation of AcrB, 1.18 Å with the L conformation of AcrB), the structure of PN2 subdomain largely differs from both the L and T conformations (local PN2 RMSD ($C_\alpha$'s): 1.83 Å with the T conformation of AcrB, 2.1 Å with the L conformation of AcrB) (Supplementary Table 2, Fig. 2G, H, and Supplementary Fig. 6A, B). Analogous to the L conformation of AcrB and AdeB, the DBP

(comprising the PN2 and PC1 subdomains) is closed in the L* conformation (Fig. 2E, F and Supplementary Figs. 5A and 6B).

Using the CAVER Analyst software[34], we identified three entry tunnels proceeding towards the (closed) DBP of the L* protomer (Fig. 2B, C). These tunnels resemble CH1–CH3 in AcrB[6,7,11]. As observed in structures of AcrB, the closure of these entry tunnels appears concomitant with the opening of an exit tunnel in the O conformation of AdeB (Fig. 2B, C). Since the DBP in the AdeB L* conformation is in a closed state (Fig. 2E, F), we conducted molecular docking calculations of ethidium and rhodamine 6G to the AdeB L*, L, and T protomers[30] and found binding of these substrates proximal to the switch loop in the AP of the L* and L protomer (Supplementary Fig. 7 and Supplementary Table 4). Top-docking poses were energy minimized to refine protein-ligand interactions, and approximate free energies of binding ($\Delta G_s$) were estimated (see Methods). The calculated $\Delta G$ values appear the highest in the DBP of the T protomer for both substrates and substantially smaller in the AP of the L and L* protomers. The L* protomer has a wider cleft between the PC1 and PC2 subdomains and the recently solved LTO AdeB trimer[30] indicated ethidium bound to the AP. Possibly, the L* protomer is the initial conformational state for binding of drugs, and drug entry might elicit clamping of the substrate (due to a PC2 subdomain movement, Supplementary Fig. 6A, B). Alternatively, the L* drug binding might induce opening of the DBP and conversion to the T state, yielding the TOO protomer with bound ethidium (AdeB-Et-I)[30].

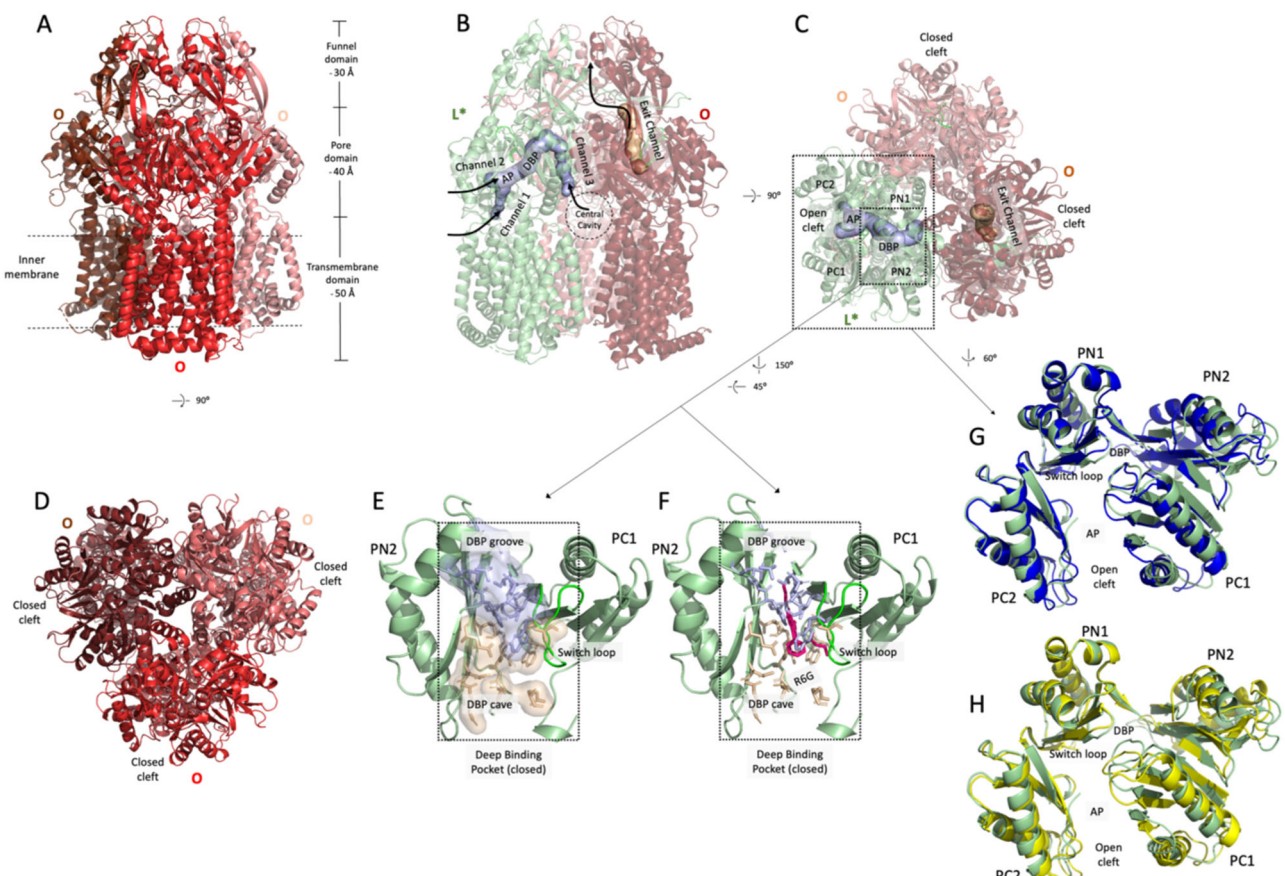

**Fig. 2 Structural models of AdeB in the OOO and L\*OO conformations. A** Side view of the C3-symmetric structure of AdeB in the OOO conformation solved at 3.54 Å. The approximate boundary of the inner membrane embedded part of AdeB is indicated by dashed lines. The dimensions of the transmembrane, porter and funnel domains are indicated. **B** From the same sample, we obtained a structure of AdeB in the L\*OO conformation at 3.84 Å resolution shown in side view. **C** Top view of AdeB in the L\*OO conformation. The L\* conformation (Chain A of the PDB entry: 7B8Q) is depicted in green, and the O conformation in light and dark red. Tunnels were calculated with CAVER Analyst[34]. Channels 1, 2 and 3, leading to the closed DBP, are marked in the L\* conformation. The PC1 and PC2 subdomains constitute an open cleft in L\* conformation (channel 2 entrance). All entry sites are closed in the O conformation, and an exit channel is opened. In the O conformation, the PC1 and PC2 subdomains are in close contact constituting a closed cleft. **D** Top view of the C3-symmetric structure of AdeB in the OOO conformation. **E** The DBP cave (wheat surface and side chains in sticks) and groove (light blue) regions are closed in the L\* conformation **F** Rhodamine 6G (R6G) derived from the co-structure of AcrBper (PDB: 5ENS), superimposed on the AdeB L\* conformation (green cartoon). Binding of R6G (pink sticks) to the DBP in this conformation is not possible. **G** A superposition on the PN1 subdomain of the L\* conformation (green cartoon) with the AcrB L (blue cartoon) and **H** the AcrB T conformation (yellow cartoon) (PDB: 4DX5) shows clear deviations in the PN2 and PC1 subdomains. Latter subdomains in L\* are arranged in a different state compared to the L and T conformations, while the PN1 and PC2 subdomains are congruent. The switch loop (L\*, green) is shifted to the inside of the DBP compared to the switch loop in the AcrB T conformation (yellow, **H**), congruent to the switch loop conformation in the L conformation of AcrB (blue, **G**). AP, access pocket; DBP, deep-binding pocket.

As the ethidium-bound structures of AdeB were published after submission of this manuscript and our efforts to obtain co-structures with novobiocin were unsuccessful, we set out to determine binding of various drugs to the open-state DBP of AcrB, for which we had previously established a successful methodology for the determination of the AcrB periplasmic part (AcrB_per) drug co-structures[18]. Based on these AcrB_per co-structures, we analyzed the activity of 20 single-substitution DBP variants of AdeB with known substrates for both AdeB and AcrB and compared these activities with DBP variants of AcrB.

**Structural investigation of substrate binding to AcrBper**. We obtained crystal co-structures of trimeric AcrBper in the LLT conformation with doxycycline (2.1 Å), fusidic acid (2.3 Å) and levofloxacin (2.7 Å) by soaking pre-grown AcrBper/DARPin crystals in the presence of doxycycline (6.25 mM), fusidic acid (5 mM) or levofloxacin (5 mM), respectively (Fig. 3 and

Supplementary Figs. 8 and 9). Doxycycline appears to bind in a congruent manner to the DBP groove as reported for minocycline[8,10,18] (Figs. 3A and 4A, B and Supplementary Figs. 8A and 9A). The carboxy amide group of doxycycline interacts with the N274 polar side chain, while F178 and F615 sandwich the aromatic ring of doxycycline. Furthermore, doxycycline is engaged in a water-mediated hydrogen bond network, extending from the 12a-hydroxyl group in doxycycline. Interestingly, we observed an additional electron density at the DBP cave of the same protomer, which we assigned to a second doxycycline molecule (DXT-2) (Figs. 3B, C and 4C and Supplementary Figs. 8A and 9A). This second binding site represents a previously undiscovered binding site for tetracycline antibiotics within the DBP, where DXT-2 interacts with the side chains of S135, F136, V139, F178, Y327, M573, F610, F615, F617 and F628. This second binding site is expected to have a lower affinity, as reflected by its higher B-factor (Supplementary Table 5). The levofloxacin binding site within the DBP overlaps substantially

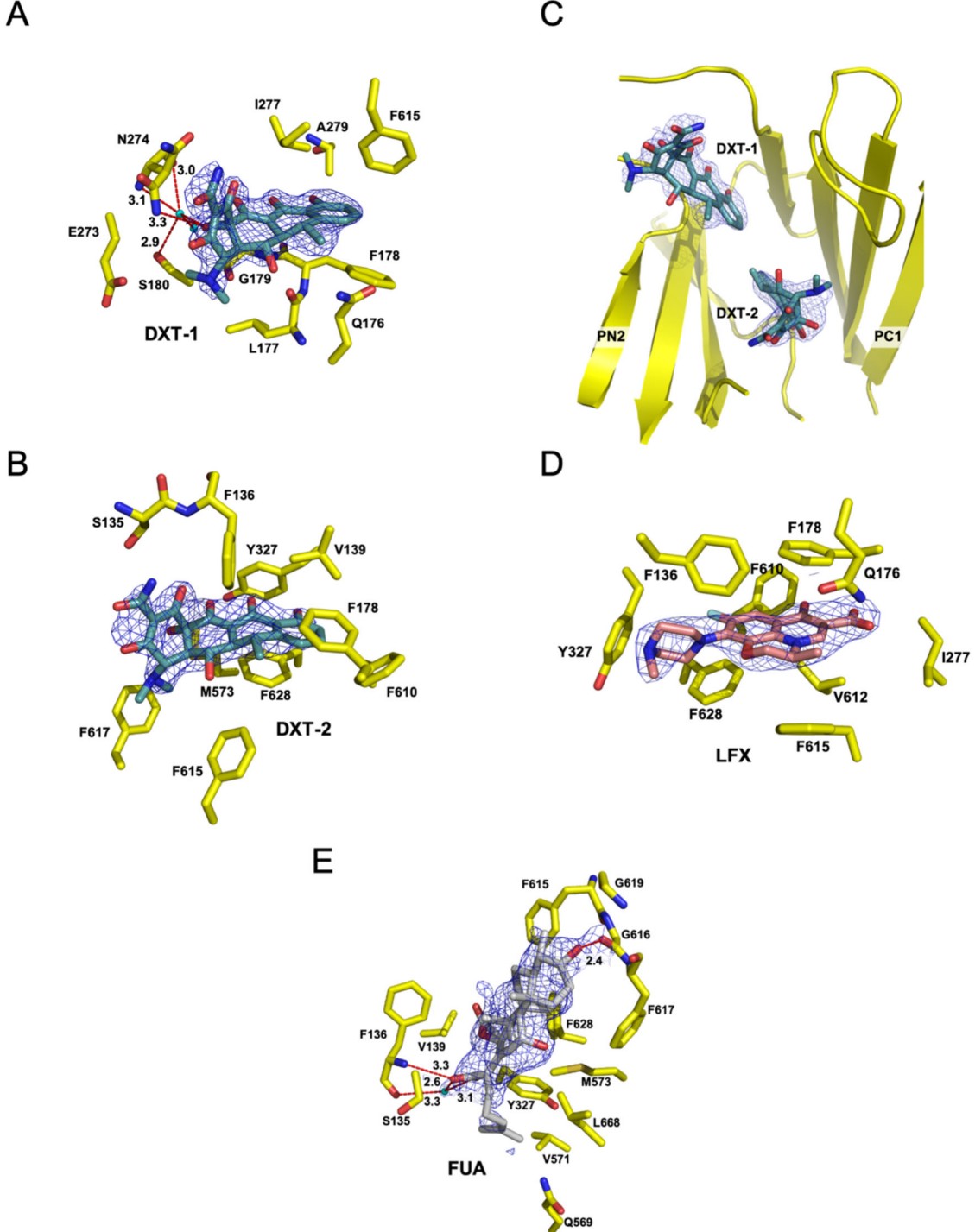

**Fig. 3 Crystal structures of doxycycline (DXT; 2.1 Å), fusidic acid (FUA; 2.3 Å) and levofloxacin (LFX; 2.7 Å) bound to the DBP of the AcrBper protomer in the T conformation. A**, **B** The DXT-soaked AcrBper crystals yielded two non-proteinaceous densities within the DBP, which we assigned to the molecules DXT-1 (**A**) and DXT-2 (**B**). While the DXT-1 binding position at the DBP groove is similar to the one previously reported for the tetracycline antibiotic minocycline[8, 10, 18], DXT-2 is located at the DBP cave region, where it interacts with the side chains of S135, F136, V139, F178, Y327, M573, F610, F615, F617 and F628. **C** Relative orientation of both DXT molecules within the DBP. **D** The LFX binding site within the DBP overlaps substantially with the binding site previously reported for rhodamine 6G (R6G)[18]. Like R6G, LFX interacts mainly with the hydrophobic side chains of F178, F610 and F628. **E** The FUA binding site resides at a more proximal part of the DBP. In addition to extended hydrophobic interactions within the binding pocket, the 3-hydroxyl group and carboxylate moiety on either side of the FUA molecule are involved in hydrogen bonding with the G616 and F136 main chains, respectively. Ligands are represented as sticks (carbon = blue-green (DXT), gray (FUA), salmon (LFX); nitrogen = blue; oxygen = red; fluoride = pale blue). The $2F_o–F_c$ electron density maps (blue-colored mesh) are contoured at 0.8σ. Residues involved in ligand binding are shown as sticks (carbon = yellow; nitrogen = blue; oxygen = red; sulfur = gold). Water molecules are represented as cyan-colored spheres. Hydrogen bonds are shown as red dashed lines, with the numbers representing the H-bond distances in Å.

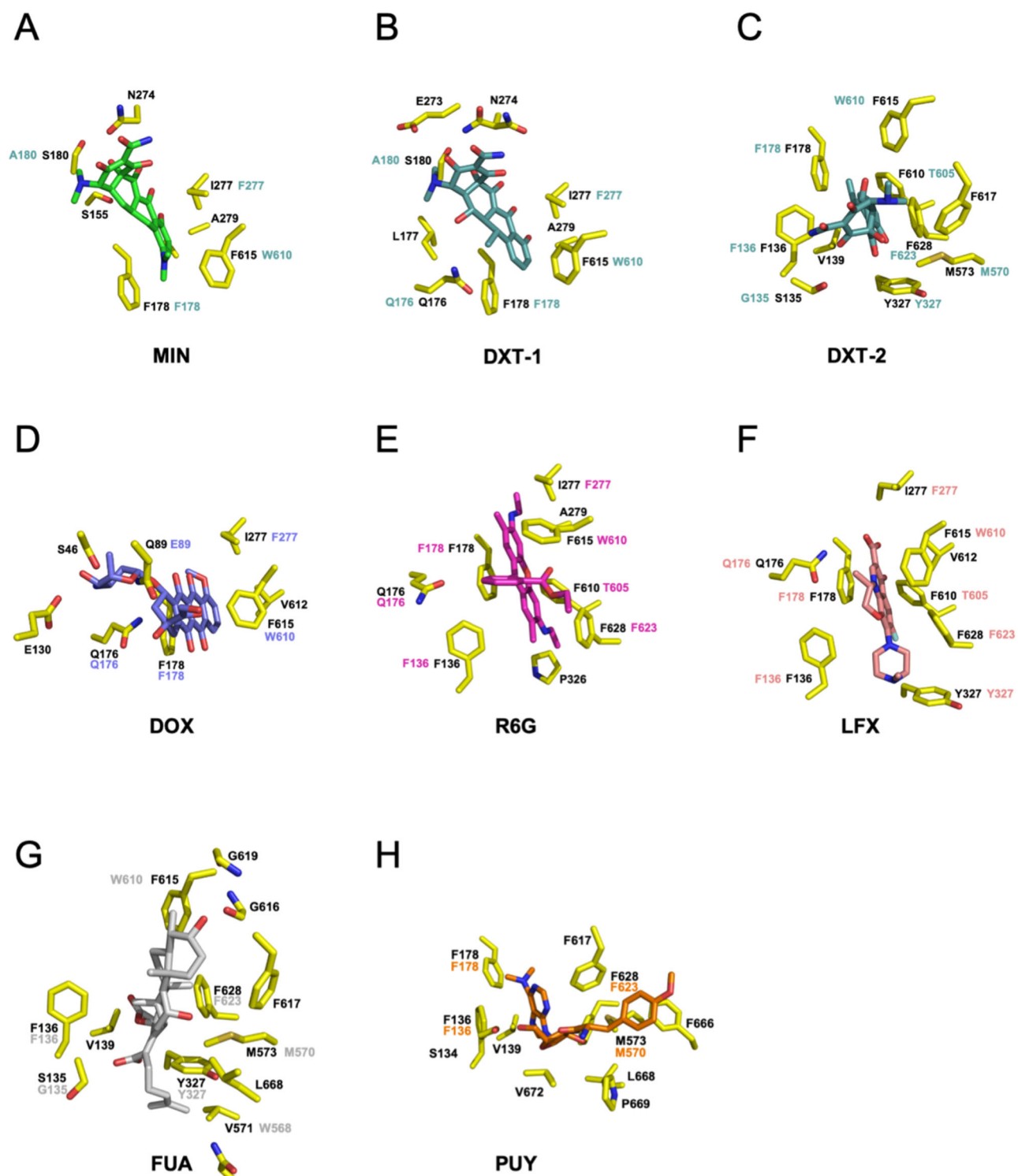

**Fig. 4 Substrate binding to the AcrB T conformation.** Binding sites within the AcrB DBP for **A** minocycline (MIN; carbon = green; PDB: 4DX5), **B** doxycycline (DXT-1) and **C** doxycycline (DXT-2); carbon = blue-green; this study), **D** doxorubicin (DOX; carbon = violet; PDB: 4DX7), **E** rhodamine 6G (R6G; carbon = pink; PDB: 5ENS), **F** levofloxacin (LFX; carbon = salmon; this study), **G** fusidic acid (FUA; carbon = gray; this study) and **H** puromycin (PUY; carbon = orange; PDB: 5NC5). AcrB residues involved in ligand binding are shown as sticks (AcrB; carbon = yellow; nitrogen = blue; oxygen = red; sulfur = gold) and by number, with colored numbers representing the corresponding AdeB residues.

with the binding site previously reported for rhodamine 6G[22] (Figs. 3D, 4E, F, 5 and Supplementary Figs. 8B and 9B). Like rhodamine 6G, levofloxacin interacts mainly with the hydrophobic side chains of F178, F610 and F628. We found fusidic acid bound to a more proximal part of the DBP, where it interacts

with the side chains of S135, F136, V139, Y327, Q569, V571, M573, F615, F617, G619, F628 and L668 (Figs. 3E and 4G and Supplementary Figs. 8C and 9C). In addition, the fusidic acid 3-hydroxyl group forms a hydrogen bond with the G616 main-chain carbonyl oxygen, while the fusidic acid carboxyl group is

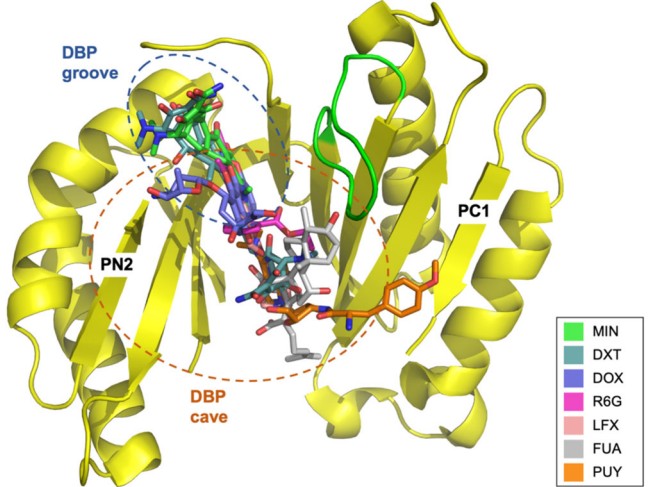

**Fig. 5 Overlay of substrates bound to the DBP of the AcrB T conformation.** Superposition of the coordinates of minocycline (MIN; carbon = green; PDB: 4DX5), doxycycline (DXT; carbon = blue-green; this study), doxorubicin (DOX; carbon = violet; PDB: 4DX7), rhodamine 6G (R6G; carbon = pink; PDB: 5ENS), levofloxacin (LFX; carbon = salmon; this study), fusidic acid (FUA; carbon = gray; this study) and puromycin (PUY; carbon = orange; PDB: 5NC5) indicate partially overlapping binding sites throughout the DBP. Substrates are shown as sticks, the AcrB PN2/PC1 subdomains are represented as yellow cartoon. The switch loop is in green.

involved in a direct interaction with the main-chain amide hydrogen and in a water-mediated hydrogen bond with the main-chain carbonyl oxygen of F136 (Figs. 3E and 4G). We considered an alternative fusidic acid conformation (Supplementary Fig. 8D; FUA (flipped)) but based on the extended hydrogen network observed in the former structure, we favor the fusidic acid conformation depicted in Fig. 3E.

**Molecular determinants of substrate binding in the DBP of AdeB.** From the AcrB/AcrBper co-structures determined in this study, we identified the interaction sites of levofloxacin, doxycycline and fusidic acid, as well as the residues involved in binding rhodamine 6G (PDB: 5ENS[18]), minocycline (PDB: 4DX5[10]) and doxorubicin (PDB: 4DX7[10]) (Figs. 3 and 4). Based on these structural insights, we constructed 20 single-substitution DBP variants of AdeB. In the first set of variants, 10 residues involved in substrate binding to AcrB/AcrBper were exchanged with Ala at homologous positions in AdeB. Six of these residues are conserved between AcrB and AdeB. A second set was generated based on non-conserved amino acids, i. e. ten residues in the DBP of AdeB were substituted with their counterparts in AcrB. As a control, we used site-directed mutagenesis on AcrB for those homolog positions, which showed in AdeB the largest drug susceptibility effects upon substitution (Supplementary Table 6 and Supplementary Fig. 10). We then tested the drug efflux capacity of the AdeB variants in drug-agar-plate dilution assays and compared it to that of wildtype AcrB (Supplementary Figs. 10, 11, 12, 13). Drug concentrations were adjusted to yield an intermediate growth of cells producing wildtype AdeB, so that both less active and hyperactive variants could be identified within one experimental setup. Results (from three independent biological repeats) were analyzed by counting the dilution steps showing cell growth. The inactive proton transport relay variant AdeB D407N was used as a negative control in this study, and the number of dilution steps was subtracted from all other dilution step numbers. Furthermore, to compare the activity with that of wildtype

AdeB, we subtracted the number of dilution steps showing cell growth for *E. coli* cells harboring wildtype AdeB (Fig. 6).

Overall, we show that all tested AcrB substrates (fusidic acid, doxorubicin, tetrapheylphosphonium, ethidium, rhodamine 6G, chloramphenicol, doxycycline, minocycline, and levofloxacin) are transported by wildtype AdeB (Fig. 6 and Supplementary Figs. 11, 12, 13). AcrB conferred in general less susceptibility than AdeB, except for tetraphenylphosphonium. Furthermore, we observed a strong tendency towards much higher AcrB-conferred resistance to all the non-polyaromatic compounds, and doxorubicin, but this effect was more moderate on minocycline and doxycycline. The overall discrepancy in resistance between AcrAB-TolC and AdeABC might be explained by the slightly lower levels of the heterologously expressed *adeB* and perhaps *adeAC* genes (Supplementary Figs. 11 and 12). However, AdeABC seems to confer higher resistance than AcrAB-TolC towards those compounds having three or more aromatic rings (ethidium, tetraphenylphosphonium, rhodamine 6G), despite the lower expression levels.

Below we compare the AdeB variants to wildtype AdeB, all of which display similar expression levels (Supplementary Figs. 11 and 12).

**Rhodamine 6G, tetraphenylphosphonium and ethidium share a binding site in AdeB.** Co-structures with ethidium and rhodamine 6G bound to the DBP of AdeB and AcrB, respectively, have been reported recently[18,30]. These molecules share characteristics of polyaromaticity (three rings or more) and an overall positive charge with tetraphenylphosphonium, for which there is no co-structure with any of the known RND transporters. In AcrB, rhodamine 6G interacts mainly with the aromatic side chains of F136, F178, Y327, and F628 (F623 in AdeB) inside the hydrophobic trap by π-π stacking, as well as with Q176 (Fig. 4E)[18]. Y327 and Q176 are conserved residues amongst RND transporters[35]. In AdeB and AcrB, Ala substitution of these residues resulted in a marked increase in rhodamine 6G susceptibility (Fig. 6 and Supplementary Table 6), which suggests a similar binding mode for this drug in the AcrB and AdeB DBP (Fig. 6, Supplementary Table 6, and Supplementary Figs. 10 and 11). The AdeB variants F136A, F178A, Y327A, Q292A and F623A appear to have a similar, but not identical, resistance phenotype with respect to rhodamine 6G, ethidium, and tetraphenylphosphonium, which suggests a shared binding site localized around these residues within the AdeB DBP. Indeed, superimposition of the AdeB T protomer-ethidium co-structure with the AcrBper-rhodamine 6G co-structure shows a common binding plane for these molecules (Supplementary Fig. 14)[18,30]. The effects of Ala substitutions are less pronounced for ethidium (except for F178A), but other substitutions appear to influence resistance to rhodamine 6G as well (E89A, Q176A, F277A).

The AdeB variant G135S has a strong lowering effect on resistance to rhodamine 6G, whereas its resistance to ethidium and tetraphenylphosphonium is hardly affected. The E151Q and W568V variants also negatively affect the resistance against rhodamine 6G, whereas there is no change in their resistance to ethidium, but these variants confer a better than wildtype resistance towards tetraphenylphosphonium.

The position of the F610 side chain in AcrB, part of the hydrophobic trap, is occupied by T605 in AdeB. Despite the important role of F610 in substrate binding/transport in AcrB[26,36,37] including binding of rhodamine 6G[18] (Supplementary Table 6 and Supplementary Fig. 14), the transport of rhodamine 6G, tetraphenylphosphonium and (to a lower extent) ethidium is disturbed in the AdeB variant T605F (Fig. 6 and Supplementary Fig. 15). By contrast, transport of levofloxacin, chloramphenicol, minocycline, doxycycline

| | FUA 2 µg/ml | DOX 8 µg/ml | R6G 60 µg/ml | ETH 60 µg/ml | TPP 250 µg/ml | CAM 1 µg/ml | MIN 1µg/ml | DXT 1 µg/ml | LFX 0.01 µg/ml | |
|---|---|---|---|---|---|---|---|---|---|---|
| AcrB | 4,00 | 3,33 | 1,00 | 1,33 | -0,67 | 2,67 | 2,00 | 2,00 | 3,67 | AcrB |
| AdeB | 0,00 | 0,00 | 0,00 | 0,00 | 0,00 | 0,00 | 0,00 | 0,00 | 0,00 | AdeB |
| E89A | 1,33 | 3,33 | -3,33 | 0,33 | 0,33 | 0,00 | 0,33 | -0,33 | 2,00 | E89A |
| F136A | -0,33 | -0,33 | -4,33 | 0,33 | -4,00 | 2,67 | 0,33 | 0,67 | 3,67 | F136A |
| Q176A | 1,00 | 0,67 | -3,67 | 0,33 | 0,33 | 1,00 | 1,00 | 0,67 | 2,00 | Q176A |
| F178A | -0,33 | -1,00 | -4,33 | -3,33 | -4,00 | -0,33 | 0,00 | -0,33 | 0,00 | F178A |
| F277A | 1,00 | 1,00 | -2,67 | 0,33 | 1,33 | 2,00 | 1,33 | 0,67 | 2,00 | F277A |
| Q292A | -0,33 | -0,33 | -4,00 | -1,00 | -1,00 | 0,00 | 0,00 | -0,33 | 0,33 | Q292A |
| Y327A | 0,00 | 1,33 | -4,33 | -1,33 | -4,00 | 2,00 | 0,00 | 0,67 | 4,00 | Y327A |
| M570A | 0,00 | 0,67 | -0,33 | -0,33 | 0,00 | 0,67 | 1,00 | 0,67 | 1,00 | M570A |
| T605A | 0,67 | 0,00 | -0,67 | -0,67 | -1,00 | -1,33 | 0,33 | -0,33 | 0,00 | T605A |
| F623A | 0,33 | -1,00 | -4,33 | -2,67 | -4,00 | -1,00 | 0,00 | 0,00 | 0,00 | F623A |
| | | | | | | | | | | |
| AcrB | 2,00 | 3,00 | 0,33 | 1,67 | -1,00 | 2,33 | 2,00 | 3,00 | 3,67 | AcrB |
| AdeB | 0,00 | 0,00 | 0,00 | 0,00 | 0,00 | 0,00 | 0,00 | 0,00 | 0,00 | AdeB |
| E89Q | 1,33 | 3,00 | 0,67 | 0,33 | 1,33 | 1,67 | 2,33 | 1,33 | 1,00 | E89Q |
| G135S | 0,00 | -0,33 | -3,00 | -0,67 | 0,33 | 1,33 | 1,33 | 0,67 | 0,67 | G135S |
| Q292K | 1,67 | 3,00 | 1,00 | 0,33 | 2,00 | 0,67 | 2,67 | 1,33 | 1,00 | Q292K |
| W568V | 1,67 | 1,33 | -1,33 | 0,33 | 1,67 | 2,33 | 2,33 | 1,67 | 2,33 | W568V |
| E151Q | 1,00 | 1,00 | -2,67 | 0,00 | 1,33 | 3,33 | 2,00 | 1,33 | 1,00 | E151Q |
| A180S | -0,33 | -1,00 | -0,67 | -0,33 | -0,33 | -0,67 | 0,67 | 0,33 | -1,00 | A180S |
| T605F | 1,00 | 0,00 | -4,33 | -1,67 | -4,00 | 3,33 | 1,67 | 1,67 | 3,00 | T605F |
| W610F | 0,67 | 1,33 | 0,00 | 0,67 | 1,33 | 0,00 | 1,33 | 1,00 | -0,33 | W610F |
| N276D | -1,33 | -1,67 | -1,67 | -1,33 | -1,67 | -1,33 | -0,33 | 0,67 | -1,33 | N276D |
| F277I | 1,33 | 1,00 | -1,33 | 1,33 | -1,67 | 3,33 | 1,67 | 1,67 | 3,67 | F277I |

**Fig. 6 Drug susceptibility profiles of E. coli harboring AdeB deep-binding pocket variants.** Analysis of plate dilution assays with E. coli AcrB, wildtype A. baumannii AdeB, inactive AcrB variant (D407N) and AdeB deep-binding pocket (DBP) variants (E89A, F136A, Q176A, F178A, F277A, Q292A, Y327A, M570A, T605A, F623A, E89Q, G135S, Q292K, W568V, E151Q, A180S, T605F, W610F, N276D, F277I). Plate dilution assays were performed with E. coli BW25113 ΔacrB ΔacrD ΔmdtBC pRSFDuetFX_MS_adeAC harboring pET24_acrB or p7XC3H_adeB_WT and mutants. Dilution series of overnight cultures with an OD$_{600nm}$ of $10^0$, $10^{-1}$, $10^{-2}$, $10^{-3}$, $10^{-4}$ and $10^{-5}$ (6 dilution steps) were spotted on Mueller-Hinton Agar plates containing 50 µg/ml kanamycin, 50 µg/ml carbenicillin and 20 µM IPTG, with or without (control plate) the tested drug. Plates were supplemented with the following compounds: 2 µg/ml fusidic acid (FUA), 8 µg/ml doxorubicin (DOX), 60 µg/ml rhodamine 6G (R6G), 60 µg/ml ethidium (ETH), 250 µg/ml tetraphenylphosphonium (TPP), 1 µg/ml chloramphenicol (CAM), 1 µg/ml minocycline (MIN), 1 µg/ml doxycycline (DXT), and 0.01 µg/ml levofloxacin (LFX). All experiments were performed in biological triplicates. The last dilution steps showing cell growth were documented and averaged (Supplementary Figs. 11–13). The numbers indicate the calculated difference to AdeB wildtype after subtraction of the negative control (D407N). Positive results (green shadings) indicate increased resistance to the drug compared to AdeB WT, negative results (red shadings) indicate decreased resistance. E. coli AcrB, wildtype A. baumannii AdeB, inactive AcrB variant (D407N) were measured twice, once in biological triplicate with the single Ala-substitutions (E89A, F136A, Q176A, F178A, F277A, Q292A, Y327A, M570A, T605A, F623A) and once in biological triplicate with E89Q, G135S, Q292K, W568V, E151Q, A180S, T605F, W610F, N276D, F277I.

and fusidic acid is better than AdeB wildtype in this variant. We suggest that in AdeB, rhodamine 6G is shifted slightly in the DBP compared to its location in AcrB (and compared to the position of ethidium (Supplementary Figs. 7 and 14)), using the extended space near T605. This might explain the severe negative effect on rhodamine 6G resistance, but less so on ethidium resistance, of cells harboring the T605F substitution. The ethylamino moiety of rhodamine 6G interacts with T605 and is slightly closer to the ethidium molecule at that side (Supplementary Fig. 7). Moreover, for ethidium, the second binding site inside the DBP might be less affected by the T605F substitution (Supplementary Fig. 7). No increase in susceptibility was observed for variant T605A, despite the observable H-bond between this residue and the rhodamine 6G ethylamino moiety from the docking analysis (Supplementary Fig. 7).

Other substitutions, like Q292K and N276D, show (in part) better than AdeB wildtype resistances (Fig. 6 and Supplementary Figs. 11, 12, 13), and appear to do so for other, non-polyaromatic compounds as well, which suggests that they elicit a general transport effect.

Of note, AdeB and AcrB activity against rhodamine 6G appears to be very sensitive to substitutions, thus resembling the characteristics of a specific binding site rather than a polyspecific binding site. Profiles of tetraphenylphosphonium and ethidium resistance indicate diverse phenotypes (Fig. 6). For ethidium, this is in stark contrast to observations made for AcrB[26,36,38], where in most cases a single-site substitution in the DBP appears not to have a strong effect on the resistance phenotype as determined by MIC measurements. AcrB substitutions F136A and F178A did

not or only mildly affect the tetraphenylphosphonium susceptibility, respectively, whereas in AdeB, these substitutions cause a strong susceptibility effect (Fig. 6 and Supplementary Table 6).

**Improved transport of levofloxacin and correlation to chloramphenicol.** Concomitant with larger susceptibilities to rhodamine 6G, ethidium and tetraphenylphosphonium caused by the F136A (except with ethidium), F277A (rhodamine 6G only), Y327A, T605F and F277I (except with ethidium) substitutions, susceptibilities to levofloxacin and chloramphenicol are, in contrast, substantially reduced (i.e., higher resistance is observed), particularly for the F136A and Y327A AdeB variants (Fig. 6 and Supplementary Figs. 11 and 12). Compared to wildtype AcrB, AcrB variants F136A and F178A do not confer any difference in susceptibility to doxorubicin[36] and chloramphenicol, and F136A does not affect levofloxacin susceptibility (Supplementary Table 6). The AcrB variant Y327A confers susceptibility to minocycline[38], rhodamine 6G, tetraphenylphosphonium, levofloxacin and chloramphenicol (Supplementary Table 6), but was reported not to change susceptibility to ethidium[38]. For AdeB, F136A causes an increase in resistance towards levofloxacin and chloramphenicol, whereas in AcrB, this substitution does not alter the susceptibility compared to wildtype AcrB. For Y327A, this contrast is even more drastic as it substantially increases the susceptibility of cells harboring the AcrB Y327A variant, whereas in the AdeB Y327A variant, susceptibilities are strongly decreased (higher resistance than wildtype). In AdeB, the contrasting effects

on susceptibilities toward levofloxacin/chloramphenicol compared to rhodamine 6G/ethidium/tetraphenylphosphonium might indicate a shared binding site for levofloxacin and chloramphenicol, as well as mutually exclusive binding of the three polyaromatic compounds rhodamine 6G, ethidium and tetraphenylphosphonium. Compared to the latter cationic polyaromatic compounds (with delocalized charge), levofloxacin and chloramphenicol comprise only one aromatic ring, and levofloxacin contains a localized negative charge (and an additional positive charge at lower pH), whereas chloramphenicol contains a local zwitterion. Despite these obvious physicochemical differences, the AcrBper/levofloxacin co-structure shows the binding of levofloxacin at the same planar level as the binding of rhodamine 6G, albeit slightly shifted towards Y327 (Fig. 4E, F). This slight deviation in orientation is also apparent compared to ethidium binding in the T protomer of AdeB (Supplementary Fig. 14B). The main interaction partners of levofloxacin seem to be the conserved AcrB residues F178 and F628 (F623 in AdeB), as well as F615 (W610 in AdeB). Removal of hydrophobic bulky residues (F136A and Y327A) in the lower part (entrance) of the DBP cave region markedly increased the levofloxacin and chloramphenicol pump activity in AdeB. Ile or Ala substitution of F277, a residue located further up in the DBP groove, also causes higher levofloxacin and chloramphenicol pump activity. Moreover, introduction of Phe in the T605F variant, also increases levofloxacin and chloramphenicol pump activity leading to higher resistance against these compounds, whereas removal of the phenyl ring of F610 in AcrB (homologous to the T605 position), causes a strong reduction in resistance against levofloxacin and chloramphenicol (Supplementary Table 6). We suggest that levofloxacin and chloramphenicol share a similar binding mode in AdeB. The binding region in AdeB appears to be different for levofloxacin and chloramphenicol compared to rhodamine 6G, ethidium and tetraphenylphosphonium. The shared binding site for levofloxacin with rhodamine 6G that is seen in the AcrB co-structures and supported by the mutational analysis data (Supplementary Table 6, except for F136A), does not seem to hold true for AdeB.

It appears that AdeB binding and transport of the tested substrates changes strongly upon removal of Phe and Tyr side chains in the DBP, and these exchanges result in different phenotypes in AcrB. For AdeB, substitutions like Q176A (which leads to a hyperactivity resistance phenotype for chloramphenicol and a severe reduction in resistance to rhodamine 6G), most likely affect the orientation(s) of the nearby F178 and F136 side chains, leading to a change in the binding properties. Similarly, G135S is expected to influence the conformational freedom of F136, leading to suboptimal binding of rhodamine 6G and slightly more favorable conditions for binding and/or transport of chloramphenicol, minocycline, doxycycline, and levofloxacin (Fig. 6 and Supplementary Figs. 12 and 14).

Minocycline, doxycycline, fusidic acid and doxorubicin, but also chloramphenicol and levofloxacin, appear to be less preferred substrates of AdeB, considering the lower susceptibility (higher resistance) effects of many of the side chain substitutions near the DBP. Ala-substitutions in general hardly affect the activities against these drugs (except for chloramphenicol and levofloxacin with higher resistances upon Ala-substitution). Almost all substitutions leading to homolog AcrB residues increase resistance to these compounds.

Exceptions are the resistance observed for these drugs by G135S (no effect on resistance to fusidic acid and doxorubicin), and A180S and N276D, which either have no effect or decrease resistance to most compounds. The AdeB T605F variant clearly confers higher resistance to cells, implying a more favorable interaction site especially for chloramphenicol and levofloxacin. Substitution of F277 with Ile or Ala resulted in a variant of AdeB

that conferred higher resistance compared to wildtype AdeB for minocycline, doxycycline, fusidic acid, doxorubicin, chloramphenicol, and levofloxacin.

The DBP of AdeB is highly lipophilic (13 non-polar residues, compared with 11 in AcrB and 4 in AcrD) with a moderate propensity for hydrogen bonding (13 side chain hydrogen bond donors/acceptors, compared with 16 in AcrB and 20 in aminoglycoside-transporting E. coli homolog AcrD). Although AdeB contains three negatively charged side chains (compared with two in AcrB and two in AcrD), it has no positively charged side chains (compared with two in AcrB and four in AcrD).

We assume that AdeB variants E89A, E89Q, E151Q, N277D and Q292K change the resistance phenotype by altering the balance of charges in the DBP. Removal of a negative charge in variants E89A, E89Q, and E151Q increases resistance to all compounds that we tested except rhodamine 6G. The additional negative charge in N277D decreases resistance to all compounds except the tetracyclines minocycline and doxycycline. Insertion of a positive charge in Q292K improves transport of all tested compounds.

In sum, it appears that AdeB displays much higher preference for polyaromatic compounds (with three or more aromatic rings), and that single substitutions changing the polyaromatic interaction sites within the DBP greatly decrease pump activity. These variants, however, transport more hydrophilic compounds, especially levofloxacin and chloramphenicol, much more efficiently. Furthermore, the lower susceptibilities conferred by many of the AdeB variants compared to wildtype AdeB suggest that levofloxacin, chloramphenicol, and other hydrophilic compounds are less efficiently transported out of the E. coli cell by the wildtype AdeB pump.

## Discussion

AdeABC is a major factor in MDR of A. baumannii, but it is tightly regulated in wildtype strains[25]. So-called "slow" porins, which have low pore-forming activity, contribute to the tight outer membrane of A. baumannii[39]. Its low permeability, in synergy with multidrug efflux pumps, is a strong defense mechanism against toxic compounds[27]. By contrast, the well-characterized E. coli AcrAB-TolC complex acts in synergy with a rather porous outer membrane[39]. Yet, porination appears to only affect permeation of a distinct set of toxins. For some hydrophobic compounds, outer membrane permeation was shown to be higher in A. baumannii compared to E. coli[40]. RND transporter complexes spanning the outer and inner membranes need to overcome different challenges in these organisms. Previous reports[19,22,27] and this study illustrate the extraordinarily broad substrate spectrum of AdeB, bestowing it with a promiscuity that is anticipated to be even higher than that observed for AcrB. Owing to the relatively fast permeation of some substrates across the more porous outer membrane, E. coli needs to react immediately. The presumed resting state of constitutively expressed AcrB is the LLL conformation[5], which has multiple channels and binding sites per trimeric complex. A current hypothesis, based on cryo-EM and in situ cryo-electron tomography studies, suggests that AcrAB diffuses along the inner membrane, transiently forming complexes with TolC. Substrate binding leads to the instant activation of the transport cycle by closing a channel formed by AcrA and opening the periplasmic α-barrel of TolC[15,16,41]. In A. baumannii, the AdeIJK efflux pump is constitutively expressed and might be comparable in its role to AcrAB-TolC in E. coli. The recent apo and eravacycline-bound structures revealed an asymmetric trimeric setup, with the protomers in three distinct states[42]. Our single-particle cryo-EM results in this work show that 90% of imaged AdeB protomers

remain inaccessible to substrates from the periplasm in the O conformation. Just 10% of analyzed protomers adopt the L* conformation. We assume that the high prevalence of particles in the OOO conformation is not an artifact, as similar trimeric conformations have been observed in another cryo-EM structure of AdeB[29] and a crystal structure of *Campylobacter jejuni* CmeB[43]. A recent study of the structure of symmetric AdeB[29] found particles only in the O conformation. Despite the O conformation containing no substrate binding sites, Su et al.[29] presented molecular docking of AdeB substrates bound to AdeB. This apparent contradiction might be explained using homology modeling of AdeB with one of the published homologous RND (co-)structures (likely in a T conformation). However, since the homology modeling procedure was neither indicated nor described as such, the impression was conveyed that the solved AdeB OOO structure included substrate binding sites. The recent report from the same lab[30] on the symmetric and asymmetric trimeric structures obtained in presence of ethidium, on the other hand, gives valuable insights. The asymmetric structures reveal multiple trimeric protomer states i.e., (i) binding/extrusion/extrusion (TOO), (ii) binding/extrusion/resting (TO-Resting), and (iii) access/binding/extrusion (LTO). The O and resting states are the only states without drug bound, and the L and T states are only present in an ethidium bound state. Whereas the L state binds one ethidium molecule in the AP, the T protomer in the TOO state binds two ethidium molecules in the DBP and one in the AP, the T protomer in the TO-Resting and LTO states, bind one ethidium molecule in the DBP and AP each.

In this report, we describe a previously undiscovered protomer conformation by single-particle cryo-EM. The L* conformation, assumed to be present in only 30% of AdeB trimers in the cell, revealed three substrate entry channels per protomer, corresponding to AcrB CH1–CH3, which lead to the (still closed) DBP. We propose that L* is the first conformation in the transport cycle of AdeB. Substrate binding via L* might lead to an induced fit conformational change (possibly to a drug-bound T conformation), priming the inner membrane pump for assembly into an active AdeABC tripartite complex. Molecular docking experiments indicate the binding of rhodamine 6G and ethidium to the L* protomer near the experimental ethidium molecule in the L state (Supplementary Fig. 7 and Supplementary Table 4). Whereas it is not possible to discern whether the L* conformation is recurring in the postulated AdeB LTO cycle, that state appears to be the only one bearing entrances from the periplasm in the absence of substrates. This would allow initial substrate entry in case drugs enter the periplasm, since the OOO state lacks substrate entry and binding sites. As the OOO and L*OO conformations and no other conformations are observed in the absence of drugs, we suggest the drug extrusion cycle starts with drug binding at the L* protomer. Binding of drugs might trigger the L* to T transition, possibly favored by the counterclockwise neighboring O protomer with a tilted PN1 subdomain. This would result in the TOO structure, as observed by Morgan et al.[30] (AdeB-Et-I, PDB: 7KGG). The other, clockwise neighboring protomer, might subsequently convert to the L state and bind substrate in the AP, leading to the LTO structure AdeB-Et-III (PDB: 7KGI)[30]. Further drug uptake and release is proposed to occur in analogy to the *E. coli* AcrB LTO transport cycle[6–8,16].

The highly reduced number of accessible binding sites together with wildtype gene repression might contribute to a tight regulation of AdeB, when the *adeABC* operon is expressed. In comparison, to date, no structures of trimeric AcrB in the OOO conformation have been reported. A trimeric arrangement with more than one O conformation was calculated to be energetically unfavorable[44] and to hypothetically induce release of AcrA, leading to a disassembly of the complex[45]. The observed OOO

conformation of AdeB in the absence of its interaction partners AdeA and AdeC is consistent with this hypothesis. In the asymmetric AdeB structure AdeB_II_Et[30], an additional, possibly intermediate state was observed, which was designated as resting state. This resting state shows structural resemblance to the previous observed C state in the MexAB-OprM tripartite structure[14]. It was suggested that this C-state precedes the O-state. It therefore appears that the single-particle Cryo-EM analysis reveals additional energy states of the RND protomers, hence conformations, which represent intermediate states between the three main L, T and O states derived from crystal structures.

For RND-type antibiotic exporters, both the existence of multiple substrate entry sites[6–8,11,12] and the variable nature of the AP and DBP contribute to the broad substrate polyspecificity observed[9,10,12]. For AcrB, four distinct substrate entry sites have been described[11,12]. However, all entry channels are thought to converge in the DBP of the T conformation, from which the substrates are expelled through the exit channel upon transition from the T to the O conformation. From superposition of the known substrate coordinates[8–10,15,18] (Figs. 4 and 5), we conclude that substrates share partially overlapping binding sites within the groove and cave regions of the entire DBP. Largely planar, heterocyclic compounds including doxycycline and levofloxacin fit into the DBP groove and upper cave region, interacting efficiently with polar side chains and hydrophobic residues of the Phe-rich area. Non-planar molecules like fusidic acid and puromycin are found at a more proximal part of the DBP. These molecules appear to bind to the AP-DBP interface, below the switch loop at a similar position as reported for erythromycin in the AP of the L conformation[9]. The co-structures with erythromycin, fusidic acid and puromycin might represent states close to the L–T transition[16,26]. In general, each co-structure can be interpreted as a snapshot of a single compound along its transport pathway. These snapshots are likely to represent local energy minima, where the substrate is bound to its preferred interaction site within the DPB. Nevertheless, upon extrusion, each drug is thought to interact with several transient binding sites[46,47]. In fact, each binding mode observed in the co-structures (Fig. 4) described above might reflect a possible position that any of the drugs can take, albeit with different affinities and hence probabilities[47]. We hypothesize that the overlapping binding sites illustrated by superposition of the different co-structures (Fig. 5) represent the transport pathway of any of the drugs during catalysis. In this way, drug substrates slide from underneath the switch loop towards the DBP, from where these drugs are actively guided upon the T–O transition through the exit tunnel.

The higher resistances conferred to *E. coli* by AdeB towards polyaromatic compounds as compared to more hydrophilic drugs and the decrease of resistance for the polyaromatic compounds upon Ala substitutions, may imply that the L* conformation, which includes three substrate transport channels (Fig. 2B), is closer to a drug binding state than might be anticipated at first sight. The reported molecular docking poses are one indication that rhodamine 6G and ethidium might initially bind in the AP of the L* protomer (Supplementary Fig. 7). Our mutagenesis analysis with AdeB indicates furthermore that a slight change in binding properties elicits differential effects, resulting in worse or much better than wildtype transport phenotypes. The effect of substitutions in AcrB, however, are either increasing the susceptibility of all drugs tested (e.g., rhodamine 6G, tetraphenylphosphonium, levofloxacin, apart from F136A for tetraphenylphosphonium and levofloxacin) or have no effect. For chloramphenicol, e.g., only Y327A and F610A increase susceptibility and the three other substitutions are without effect. In contrast, substitutions like F136A and Y327A in AdeB greatly *increase* the resistance towards levofloxacin and chloramphenicol.

Of interest, all T protomers shown by Morgan et al.[30], contain next to the binding of one or two ethidium molecules in the DBP, always an ethidium molecule in the AP. This contrasts with AcrB, where the DBP has been found open in the T state even in the absence of substrates[6,8] and thus far has not shown to be binding drugs to both AP and DBP in the same protomer. We speculate that the binding of drugs to the AcrB DBP might occur via conformational selection. For AdeB, the mechanism might be rather an induced fit. The AP of the L* protomer is occupied by the drug, which might induce the opening of the DBP. Polyaromatic substrates might be acting subsequently more effective with the aromatic side chains of the AdeB DBP and allow for tight interactions. By contrast, the DBP might be less suitable for compounds with fewer or no aromatic rings that bind primarily by hydrogen bonding, as seen for levofloxacin, chloramphenicol, minocycline, and doxycycline. The mutagenesis study described here appears to suggest that removal of one of the bulky Phe, Tyr or Trp residues results in a general widening of the binding pocket, creating a less tight hydrophobic environment and the possibility for compounds like levofloxacin, chloramphenicol, minocycline, and doxycycline to interact more flexibly with the remaining exposed side and main-chain hydrogen bonding donors and acceptors. The L* state might be a state only present in the absence of drugs in the L*OO conformation, next to the OOO conformation. Once drugs are present, the other protomers like L and T with drug bound might influence the conformation of their neighboring partners within the trimer. The TOO state (AdeB-I-Et) observed by Morgan et al.[30] might represent likewise an initial state prior to the LTO state (AdeB-III-Et).

This report indicates that substrate binding to RND transporters cannot be understood only by substitution-based functional analysis, nor exclusively using co-structures. Indeed, much more structural information is needed to understand AdeB (and AcrB) substrate binding, including information on other conformational states, which could possibly be inferred from analyses of inhibitor binding or substitution variants, complemented by in silico studies. Moreover, the insights from one specific RND transporter cannot easily be extrapolated to homologous transporters. To understand the efflux phenotype of pumps in their cognate environments, these must be studied in their native hosts, as the synergy between OM permeability and efflux pumps has been evolutionary developed.

## Methods

**Cloning of *adeB*, *adeA* and *adeC*, site-directed mutagenesis of *adeB*.** The genes encoding AdeA (ABAYE1821), AdeB (ABAYE1822) and AdeC (ABAYE1823) were PCR-amplified from the genomic DNA of *A. baumannii* AYE. The *adeB* gene was cloned into pINIT_cat via FX_cloning[48]. All *adeB* mutants were constructed by site-directed mutagenesis using inverse PCR with the template pINIT_*adeB*. For further experiments, wildtype *adeB* (WT) and the resulting mutants were subcloned by FX-cloning into p7XC3H[48]. The genes *adeA* and *adeC* were inserted into the plasmid pRSFDuetFX_MS, a modified version of pRSFDuet-1 (Novagen, Merck, Germany) (Supplementary Fig. 16). First, *adeA* was cloned into pINIT_cat by FX-cloning then subcloned into pRSFDuetFX_MS with the same method. The *adeC* gene was inserted into pRSFDuetFX_MS using the restriction enzymes KpnI and PacI. The identities of *adeB-His₁₀* (WT and mutants), *adeA-Myc* and *adeC-Strep* were confirmed by Sanger sequencing. All primers are listed in Supplementary Table 7.

**Expression and purification of AdeB.** *E. coli* C43(DE3) Δ*acrAB*[12] harboring p7XC3H_*adeB* was grown in TB-medium (12 g/l tryptone, 24 g/l yeast extract, 5 g/l glycerol, 2.31 g/l KH₂PO₄, 12.5 g/l K₂HPO₄) with 50 µg/ml kanamycin to an OD₆₀₀ of 1.2. Protein expression was induced with 0.5 mM IPTG and proceeded overnight at 20 °C. Cells were harvested and resuspended at 2 ml/g wet cells in 20 mM Tris-HCl pH 8.5, 500 mM NaCl, 2 mM MgCl₂ supplemented with 10 µg/ml lysozyme, 10 µg/ml DNase and 200 µM PMSF. The cell suspension was stirred for 1 h and then lysed by two runs through Stansted SPCH-EP-10 pressure cell homogenizer (Homogenizing Systems Ltd, UK) at 14–29 kPsi. The lysate was then centrifuged in two steps, first for 45 min at 16,264 × *g* (Sorvall GSA rotor) to remove cell debris, then for 1 h at 186,010 × *g* (Beckmann 45 Ti rotor) to isolate the cell membranes. The membrane pellet was resuspended at 4 ml/g wet weight membranes in 20 mM Tris-HCl pH 8.5, 500 mM NaCl, then diluted in 8 ml/g 20 mM Tris-HCl pH 8.5, 150 mM NaCl, 10% glycerol. The mix was supplemented with 10 mM imidazole and 1 % *n*-Dodecyl-β-D-maltopyranoside (DDM). Membrane proteins were solubilized for 1.5 h. After centrifugation of the mixture for 1 h at 186,010 × *g* (Beckmann 45 Ti rotor), the supernatant was incubated with Ni-NTA (Qiagen, 0.2 ml of 50% slurry per ml supernatant) for 1 h. The loaded resin was washed consecutively with 20 mM Tris-HCl pH 8.5, 150 mM NaCl, 10% glycerol supplemented with 0.03% DDM and (1) 10 mM, (2) 30 mM, (3) 50 mM and (4) 100 mM imidazole, respectively. Washing steps 1 and 2 were performed with 15 bed volumes of buffer, steps 3 and 4 with 5 bed volumes of buffer. AdeB was eluted with 5 bed volumes of the same buffer containing 300 mM imidazole. Purified AdeB was concentrated with Amicon 100 Ultra-15, 100 kDa cutoff to 1 ml and applied to SEC using a Superose 6 10/300 increase column equilibrated with 20 mM Tris-HCl pH 8.5, 150 mM NaCl, 0.03% DDM. All purification steps were performed at 4 °C.

**Reconstitution of AdeB into salipro nanodiscs.** The scaffold protein SapA was expressed and purified as previously described[28] with some modifications. All growth media were supplemented with 34 µg/ml chloramphenicol and 50 µg/ml kanamycin, but no tetracycline was added. Cultures were incubated overnight at 20 °C after induction and cells were lysed by two passages through Stansted SPCH-EP-10 pressure cell homogenizer (Homogenizing Systems Ltd, UK) at 21 kPsi. In addition, imidazole concentrations during IMAC were reduced to 20 mM in the washing buffer and 100 mM in the elution buffer; the lysis buffer was free of imidazole. For the reconstitution of AdeB, a protocol adapted from Du et al.[49] was used. Purified SapA and *E. coli* polar lipids were mixed in a molar ratio of 1:10 and diluted to a final volume of 500 µl with 50 mM sodium acetate pH 4.8. The mix was incubated for 10 min at 37 °C. Insoluble material was removed by centrifugation for 10 min at 20,000 × *g*. The supernatant was applied to a desalting column for buffer exchange to 20 mM Tris-HCl pH 8.5, 150 mM NaCl. Purified AdeB was added to the mixture and the volume was filled up to 1 ml with the same buffer. At this step, the final molar ratio of AdeB:SapA:lipid was adjusted to 1:10:100. The mix was incubated for 30 min and then dialyzed overnight against 500 ml of 20 mM Tris-HCl, pH 8.5, 150 mM NaCl. After exchanging to fresh buffer, the sample was dialyzed for an additional 3 h. The reconstitution mix was concentrated and purified by SEC using a Superose 6 10/300 increase column equilibrated with the same buffer. After the addition of AdeB to the reconstitution mix, all steps were performed at 4 °C.

**Cryo-EM sample preparation and data collection.** For the structural analysis of AdeB by cryo-EM, two datasets from consecutive purifications were recorded. For the first dataset, purified AdeB Salipro particles were concentrated to 0.6 mg/ml. For the second dataset, AdeB Salipro particles concentrated to 0.68 mg/ml were preincubated with 1 mM novobiocin for 1 h on ice. 3.5 µl of the samples were applied to glow discharged Quantifoil R1.2/1.3, 300-mesh Cu holey carbon grids (Quantifoil) and vitrified in liquid ethane using a Vitrobot (FEI) at 100% humidity and 4 °C. The blotting paper (grade 595; Whatman) was equilibrated for 1 h in the machine, and blotting force and time were set to –25 and 6 s, respectively. Grids were transferred to a Titan Krios (FEI) operating at 300 kV. The microscope was installed with a K2 summit direct detector (Gatan) and a postcolumn energy filter (GIF Quantum, Gatan) set to a slit width of 20 eV. Micrograph stacks of 48 images were recorded in counting mode using Serial-EM at a magnification 130,000 x (pixel size of 1.05 Å) with a defocus of −1 to −3.5 µm (dataset 1) and −1.5 to −4.0 µm (dataset 2). The acquisition time, dose rate, and total dose for a single micrograph were 10.6 s, 5.65 (e⁻/Å²)/s, and 60 e⁻/Å² for dataset 1; 8.16 s, 7.45 (e⁻/Å²)/s, and 60.79 e⁻/Å² for dataset 2.

**Cryo-EM data processing.** A total of 1997 micrograph stacks from both datasets were aligned with UCSF MotionCor2[50]. The contrast transfer function (CTF) parameters were estimated using Gctf[51]. Further processing steps were performed with Relion 3.0[52]. A small set of approximately 1000 particles was picked manually, 2D classes were generated and models for automated picking were selected. We extracted 381,635 particles after autopicking. Poor-quality particles were removed after each of several iterative 2D classification rounds. We used 192,349 particles as an input for an unsupervised 3D classification with a low-pass filtered reference map of trimeric *E. coli* AcrB (pdb 2GIF)[6]. Correction of per-particle defocus and beam-induced motion was performed with 132,326 selected particles. After a high-resolution auto-refinement with applied C3 symmetry, a density map of AdeB in OOO conformation was obtained at 3.54 Å global resolution. The same set of particles was expanded based on their C3-symmetry, which resulted in triple the number of particles rotated along their symmetry axis. Two monomers were subtracted from each particle, resulting in 397,038 monomer particles. An unsupervised 3D classification was performed using a soft monomer mask and C3 symmetric AdeB as a reference map. The classification was performed without image alignment and the regularization parameter T was set to 15. In a total of 10 classes, one homogenous class comprising 35,170 particles showed an altered conformation. The class was selected for a high-resolution auto-refinement without applied symmetry, which resulted in the density map of the AdeB L* conformation

at 3.95 Å global resolution. The same particles were trimerized again by reverting the subtraction. Then, 280 duplicated particles were removed and a high-resolution auto-refinement without applied symmetry was performed. With this, a density map of AdeB in L*OO conformation at 3.84 Å resolution was obtained. All final refinement steps were performed using solvent flattened Fourier shell correlations (FSC) and a soft mask created from a previous refinement. Global resolution values are based on the gold standard Fourier shell correlation (FSC). Local resolution estimations were performed for all maps using the Local resolution implementation of Relion 3.0[52]. An initial structure of AdeB was generated with Phyre2[53], using the T (for AdeB L*) and O (for AdeB O) conformations from PDB 4DX5[10] as a starting model. In a Phenix real space refinement, the model was adapted to the calculated densities. The resulting structure was optimized using Coot[54] and validated with Molprobity[55].

**AcrBper expression, purification, and crystallization**. Overexpression, purification and crystallization of AcrBper and DARPin clone 1108_19 were performed as previously described[7,18], with minor changes. In brief, purified AcrBper and DARPin were mixed in a molar ratio of 1:1. AcrBper/DARPin crystals were grown at 18 °C in hanging drops containing 1.5 μl AcrBper/DARPin solution (total protein concentration of 15 mg/ml) and 1.5 μl reservoir solution over 800 μl precipitant solution (0.1 M MES pH 6.5, 0.21 M NaCl, 11.5% PEG 4000) in the reservoir well.

**Soaking of AcrBper/DARPin crystals with doxycycline (DXT), fusidic acid (FUA), and levofloxacin (LFX)**. Solutions of doxycycline (50 mM), fusidic acid (19 mM) and levofloxacin (19 mM) were prepared in AcrBper purification buffer (10 mM HEPES pH 7, 150 mM NaCl). These stocks were diluted with reservoir solution to yield a 6.25 mM (doxycycline) or 5 mM (fusidic acid, levofloxacin) soaking solution. Pre-grown AcrBper/DARPin crystals were transferred into 1 μl hanging drops of these soaking solutions and incubated over 800 μl reservoir solution for one week at 18 °C. For cryo-protection, the crystals were briefly soaked in a solution containing 15–25% PEG 200/300 (and the respective substrate) in reservoir solution.

**X-ray data collection and structure determination**. X-ray data were collected at the beamlines P13 and P14 (Deutsches Elektronen Synchrotron, Hamburg, Germany) and PX1 (Synchrotron SOLEIL, Paris, France). Datasets were processed using XDS[56] and AIMLESS from the CCP4 suite[57]. The same set of R-free reflections (5% of the AcrBper apo dataset) was used for all datasets. Phases of the ligand bound AcrBper/DARPin structures were derived from Rigid body refinement (REFMAC5[58] or phenix.refine from the PHENIX package[59]) using the AcrBper/DARPin apo coordinates (PDB ID code 5EN5[18]) as an input model. Model building was done using Coot[54], followed by restrained refinement using REFMAC5 or phenix.refine. Descriptions for the assigned ligands (doxycycline, fusidic acid and levofloxacin) were taken from the COOT monomer dictionary. Ligand (doxycycline and levofloxacin) occupancies were refined using the respective option in phenix.refine. Structure validation was accompanied by data quality analysis using MolProbity[55]. Polder maps were generated using Polder Maps[60] from the PHENIX package. Statistics from data processing and refinement are listed in Supplementary Table 5. LigPlots were generated using LigPlot+[61]. Figures were prepared using PyMOL (www.pymol.org).

**Drug-agar-plate dilution assays in E. coli**. To test the drug efflux capability of different AdeB and AcrB variants, agar-plate dilution assays were performed as described previously[62], with some modifications. E. coli BW25113(DE3) ΔacrB ΔacrD ΔmdtBC harboring pRSFDFX_MS_adeA_adeC together with pET24_acrB_His[63], p7XC3H_adeB WT, inactive mutants D407N or deep-binding pocket mutants were grown overnight at 37 °C. Dilution series were prepared starting from $OD_{600}$ $10^0$ to $10^{-5}$ in 10-fold dilution steps. 4 μl drops were spotted on Mueller-Hinton agar plates supplemented with 50 μg/ml Kanamycin, 50 μg/ml carbenicillin, 20 μM IPTG and the tested drug. Plates were incubated overnight at 37 °C. As a control, the same experiment was performed on plates without kanamycin and carbenicillin. Expression levels were determined from overnight cultures grown in medium containing 50 μg/ml kanamycin, 50 μg/ml carbenicillin and 20 μM IPTG. After harvesting of cells, lysis was performed by shearing force with Fastprep-24 (MP) and samples were solubilized in 1% DDM for 1 h. After centrifugation for 10 min at 191,531 × g in a TLA 100 rotor, AcrB-His, AdeB-His (WT and variants), AdeA-Myc and AdeC-Strep were detected in a Western blot by their respective tags. For detection of AcrB (WT and variants) an anti-AcrB antibody[12] (1:5000) was used.

**Ethidium accumulation assay**. E. coli BW25113(DE3) ΔacrB ΔacrD ΔmdtBC harboring pRSFDFX_MS_adeA_adeC together with pET24_acrB_His, p7XC3H_adeB WT, inactive mutant D407N, T605F or N276D were grown to an $OD_{600}$ of 0.5–0.6 at 37 °C, then induced with 20 μM IPTG. After 1.5 h, cells were harvested, washed with 20 mM potassium phosphate phosphate pH 7.0, 1 mM $MgSO_4$, 0.2% glucose and resuspended in the same buffer. The cell suspension was adjusted to an $OD_{600}$ of 2. After the addition of ethidium bromide in a final concentration of 20 μM, the accumulation was observed with Tecan Microplate Reader (Tecan, Switzerland) at excitation and emission wavelengths of 535 and 610 nm, respectively. Expression levels in the same cell suspensions were analyzed as described in the previous section.

**Molecular docking**. Molecular docking calculations targeting AdeB were performed using two different packages, namely AutoDock VINA[64], and the recently developed GNINA program[65], whose scoring function is based on convolution neural networks. Re-docking of ethidium onto the structure of AdeB with PDB_ID: 7KGI was successful using both codes (Supplementary Table 4). For consistency, the same input files, structures, and settings (in particular, the same box center and dimensions) were employed with both the docking programs. Default settings were used in all cases, except for the exhaustiveness parameter (giving a measure of the exhaustiveness of the local search), that was set to 512 (default 8). Protein and ligand input files in PDBQT format were prepared with AutoDock Tools[66]. Protein flexibility was considered indirectly by employing five different conformations of AdeB. Namely, the R protomer in 7KGH, the T protomer in 7KGH and 7KGG, and the L and T protomers in 7KGI were aligned to the L* monomer in 7B8Q, also included in the pool of receptor structures. Ligand input configurations were obtained from quantum/mechanical optimizations at the density functional theory level (B3LYP/6-31G**) in implicit solvent, as detailed in Malloci et al.[67]. The two compounds were considered flexible during docking (the number of rotatable bonds being 4 and 8 for ethidium and rhodamine 6G, respectively). Two sets of guided docking runs were performed for all ligands using two rectangular boxes of dimensions $30 \times 30 \times 30$ Å$^3$ partly overlapping and centered at two different sites: the first one at the center of mass of ethidium bound in the AP of the L monomer in 7KGI and the other one at the center of mass of the two molecules of ethidium bound in the DBP of the T monomer in PDB_ID 7KGG (Supplementary Fig. 17). For each program employed, each receptor structure, and each docking site, the top 10 docking poses were retained, totaling to 200 modes of binding per compound (10 poses x 5 AdeB structures x 2 programs x 2 sites). Top-scoring poses were energy-optimized to refine protein-ligand interactions using the molecular dynamics package AMBER18[68] and the protocol detailed in Basciu et al.[69]. For each optimized binding mode of both ethidium and rhodamine 6G, the (pseudo) free-energy of binding ΔG was evaluated as detailed in previous work[46] using the Molecular Mechanics–Generalized Born Surface Area (MM-GBSA) method[70].

**Reporting summary**. Further information on research design is available in the Nature Research Reporting Summary linked to this article.

## Data availability
Atomic coordinates and structure factors reported in this paper have been deposited in the Protein Data Bank under accession numbers 7B8P (AdeB-OOO), 7B8Q (AdeB-L*OO), 7B8R (AcrBper/DARPin in complex with Doxycycline), 7B8S (AcrBper/DARPin in complex with fusidic acid), 7B8T (AcrBper/DARPin in complex with Levofloxacin). Atomic coordinates that were used and support the findings of this study are available in the Protein Data Bank under accession numbers 5ENS, 7KGI, 7KGH, 7KGG, 7KGD, 4DX5, 4DX7, 5NC5, 6OWS, 6IIA, 5LQ3, 4MT1, 3K07, 3K0I. Source data for Supplementary Figs. 1, 3, 4, 10-13 and 15 are provided with this paper. Source data are provided with this paper.

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

## Acknowledgements

This work was supported by the German Research Foundation (SFB 807, Transport and Communication across Biological Membranes), the DFG-EXC115 (Cluster of Excellence Frankfurt-Macromolecular Complexes) to K.M.P., and by the DFG FR-1653/12 and SFB902 (B5) to A.S.F. The research leading to these results was conducted as part of the Translocation consortium (www.translocation.eu) and has received support from the Innovative Medicines Joint Undertaking under Grant Agreement no. 115525, resources which are composed of financial contribution from the European Union seventh framework program (FP7/2007-2013). A.V.V. and G.M. received support from the National Institutes of Allergy and Infectious Diseases project number AI136799. We thank the beamline staff at SOLEIL Synchrotron in Saint Aubin, France (Proxima-1, Proposal Number: 20140708, 20140860, 20150100), and Deutsches Elektronen Synchrotron (DESY) in Hamburg, Germany (P13, P14, Proposal MX317, MX584) for their excellent support. A.V.V. and G.M. thank Paolo Ruggerone (University of Cagliari) for precious suggestions and useful discussions.

## Author contributions

A.O.C., J.W., H.S., A.S.F. and K.M.P. designed experiments. A.O.C. and J.W. performed AdeB production, membrane preparation, and protein purification. A.O.C. performed reconstitution and cryo-EM sample preparation. A.O.C. and A.S. collected cryo-EM data. A.O.C. and J.R. processed cryo-EM data and solved the structure of AdeB. J.W. and H.S. performed AcrBper and DARPin production, protein purification and crystallization, collected synchrotron data and determined crystal structures. A.O.C. (AdeB/AcrB substitutions), J.W. (AdeB/Ala substitutions), and J.K. (AcrB/Ala substitutions) performed biochemical/microbiological assays and western blot experiments. A.V.V. and G.M. conducted docking, molecular mechanics, and free-energy calculations. A.O.C., J.W., J.K., A.S.F. and K.M.P. contributed to manuscript preparation.

## Funding

## Competing interests

The authors declare no competing interests.
