## [Peer Review File · Nature Communications]

REVIEWER COMMENTS

Reviewer #1 (Remarks to the Author):

The authors solved the structure of AdeB, but the substrate binding mechanism is speculated by the co-structures of AcrB with various substrates and mutagenesis and the functional test. This makes the cryo-EM structure of AdeB less critical in the manuscript. In the discussion part, I think the authors discussed the substrate binding specificity and substrate transport pathway by superimposing different substrate structures.

Nevertheless, as the author mentioned, "The DBP of AdeB is highly lipophilic (13 non-polar residues, compared with 11 in AcrB and 4 in AcrD)...", the DBP is not necessarily conserved. This could explain why mutagenesis-based drug resistance behaved differently between AcrB and AdeB, and it is not surprising and makes the deduction from AcrB to AdeB not straightforward. For example, as the author claimed, "We suggest that in AdeB, R6G is shifted slightly in the DBP compared to its location in AcrB...", the binding mechanism for R6G is still inconclusive given the different performance in AcrB and AdeB.

The corresponding residues could be located by sequence alignment, making the newly solved "OOO" and "LOO" structures of AdeB is not necessarily important compared to previous published AdeB structures. It will help readers understand it by directly comparing the pore-forming three alpha-helices that connect the funnel in the middle. A statement of conformation is "in between "L" and "O" states" need to be more specific to demonstrate the comparison between this new structure with previous determined models both AdeB and AcrB.

The mechanism could be better demonstrated if the author could make more use of the structure of AdeB, like

1. Biophysical experiments applied to AdeB would be needed to investigate/verify the binding mechanism proposed by the authors.
2. Molecular dynamics or docking, based on the AdeB structure, could be conducted to verify the binding mode of at least one antibiotic.

Besides, there is some minor change required for revision:

1. Figure 2E is too mass in the background, needs to be replaced with a cleaner version.
2. In Figure 2F, color make the reader not easy to recognize. The author needs to reconsider the color code used in the figures.
3. In Figure 3 and Figure 4, the name of each substrate could be labeled on the figure to help reading.

Reviewer #2 (Remarks to the Author):

The paper entitled « Structural and functional analysis of the promiscuous AcrB and AdeB efflux pumps suggests different drug binding mechanisms » by Alina Ornik-Cha, Julia Wilhelm et al. reports structures of two homologous RND pumps, AcrB from the efflux pump AcrAB-TolC efflux pump in *E. coli* and AdeB from the AdeABC efflux pump of *A. baumannii*. The structures were respectively obtained by X-ray crystallography (in that case the protein was detergent-solubilized) and cryo-EM (after reconstitution of the protein into Salipro nanodiscs, providing a native-like lipid environment to the protein). In that study, authors also analyze the binding modes of AdeB and AcrB of nine substrates in light of three new crystal co-structures of AcrB periplasmic domain in complex with levofloxacin, doxycycline and fusidic acid.

The authors reach two main findings:

AdeB adopts two conformations in the conditions of the experiments: most of the particles are described in a resting state with all protomers in the O conformation (« OOO ») and a subpopulation of particles (30%) adopting a so-called L*OO conformation, in which one of the protomers is in a transition state between the L and T conformations, hence its denomination « L* ».

Different, previously unknown, binding modes are described for key residues in the binding of structurally diverse drug compounds thanks to *in vivo* activity assays.

The manuscript is well and clearly written and experiments are well designed. Results are convincing and novel. Results and discussion should also be of interest and beyond the efflux pump field, in particular, the discussion regarding binding properties and related conformational effects (conformational selection versus induced fit) is of general interest.

Hence we strongly encourage the editor to accept this paper for publication provided that the following comments and corrections are addressed.

lane 26: it is stated there that 10% of the protomers adopt the intermediate state. In the text (lane 152), it is written that 30% of the particles adopt the intermediate state. Of course, both statements are in perfect accordance but it would be clearer for the reader to adopt a single formulation.

lane 140: the authors report the structure of AdeB. Have they considered co-reconstituting AdeA ?

lane 144: several tools are available for the stabilization and manipulation of membrane proteins for cryoEM studies. What was the specific advantage of the salipro methodology compared to SMALPs or amphipols ?

lane 152: it is written in the Material and Methods section that two datasets were used to obtain the EM structure. The second one contained novobiocin. Did the presence of novobiocin affect the proportion of L*OO ?

lane 154: It was recently shown that MexB, from the MexAB OprM of *Pseudomonas aeruginosa*, adopts a new asymmetric conformation where the exit pathway of the protomer involved in drug release is closed (dubbed « Closed conformation ») : ref: Glavier, M. et al. 2020 Nat Commun 11, 4948.

To what extent the L* state described here from AdeB compares to the C state described in MexB ?

lane 157 to 163: RMSDs should be stated in Å.

lane 181: at what cutoff are the EM maps displayed ?

manipulating the structures with Chimera, we find that:

_ chain a of 7B8P perfectly aligns with chain c of 7B8Q

_ chain c of 7B8P aligns with chain b of 7B8Q

_ chain b of 7B8P aligns with chain a of 7B8Q with significant RMSD.

Do the authors confirm that chain a of 7B8Q corresponds to the L* ? It would be interesting to add these information in the manuscript.

lane 386: it would have been interesting to compare the effects of aminoglycosides. Did the authors consider doing so ?

lane 493: it is described that the DBP of AdeB remains in the L* conformation in the absence of substrate although it is found open for AcrB in the same condition. Do the authors suggest that the L* conformation does not exist in AcrB ?

lane 544 / lane 550 / lane 551: volumes are set as proportional to « grams ». We guess that is is per gram of wet cell pellet ...

lane 548 versus lane 569: pressures are defined in kPsi (lane 548) or bar (lane 569). please correct.

lane 642: at what temperature is soaking performed ?

lane 666: correct « Müller Hinton » for « Mueller Hinton ».

lane 924, Figure S7B: do the authors know why AdeC displays two bands ?

lane 997: there seems to be a typo for construct #50.

Reviewer #3 (Remarks to the Author):

This manuscript reports structure-functional analyses of AdeB from *Acinetobacter baumannii*. The structure of AdeB was solved using cryo-EM and twenty mutants in putative binding sites of AdeB were constructed. The manuscript also reports the ligand-bound structures of the periplasmic domain of AcrB from *E.coli*. These structures of AcrB are used as guides in selection of amino acid residues of AdeB for mutagenesis. The major conclusion is that binding sites of AdeB are different from those of AcrB. *A. baumannii* and its AdeB are important targets for drug discovery and the reported findings provide new insight into role of specific amino acid residues of AdeB in the ability of this transporters to reduce inhibitory concentrations of antibacterial agents. The manuscript is clear and concise. However, there are major deficiencies that need to be addressed.

1. The impact of AdeB mutations is analyzed in the recombinant *E.coli* cells lacking *acrB*. The differences between *E.coli* and *A.baumannii* are substantial and the same mutants could have different phenotypes depending on the host. It is critical that at least some of the mutants are tested in *A. baumannii* background, see also below.

2. The properties of *A. baumannii* outer membrane are indeed quite different from those of *E.coli*. The authors emphasized the difference between porins but ignored the differences in the structures of lipid A in LPS. *A. baumannii* outer membrane is more permeable than that of *E.coli* to hydrophobic compounds, because *A. baumannii* utilizes fatty acids for growth. *E.coli* porins make its outer membrane more permeable than *A. baumannii* for hydrophilic molecules, but it is significantly less permeable than *A. baumannii* to hydrophobic compounds. Analyses of antibacterial activities in *E.coli* bias efflux pumps and they appear to be more efficient against hydrophobic molecules. Most of the authors interpretations about properties of substrates are biased by their permeation across the outer membrane. This is another reason why mutants should be analyzed in the context of the native host.

3. Authors devote significant attention to aminoglycosides both in Introduction and in Discussion. Indeed, resistance to aminoglycosides is one of the signature features of AdeB. However earlier studies (see Leus et al., 2018) showed that AdeB overproduction is required but not sufficient for aminoglycoside resistance of *A. baumannii*. Authors do not report the results with aminoglycosides. But these are important. The speculative statements in Discussion about possible alternative roles of AdeB in aminoglycoside resistance need to be substantiated by data.

4. The structures of AcrB are not integral to the presented results and their inclusion into this study is somewhat awkward. The sites are different between AcrB and AdeB. Ligand-bound structures of AdeB would be more revealing. Inclusion of AcrB mutants, see below, could make the study more cohesive.

5. Comparison of transporters' activities in *E. coli* is very unreliable, because of differences in the expression levels between native and recombinant AdeABC. Most of the AcrB residues in binding sites were already analyzed in previous studies. Inclusion of at least some of AcrB variants with mutations in analogous positions could make the comparison in *E. coli* host more reliable.

6. MIC measurements are needed to make the data more compatible with previous studies.

REVIEWER COMMENTS

Reviewer #1 (Remarks to the Author):

The authors solved the structure of AdeB, but the substrate binding mechanism is speculated by the co-structures of AcrB with various substrates and mutagenesis and the functional test. This makes the cryo-EM structure of AdeB less critical in the manuscript. In the discussion part, I think the authors discussed the substrate binding specificity and substrate transport pathway by superimposing different substrate structures.

We like to thank the reviewer for the in-depth review of our work. We suggest from the Cryo-EM structure of AdeB that the transport of drugs is different from that postulated for AcrB. The AdeB L*OO conformation is a state not found in AcrB (usually LTO, but other combination like LLT, LTT, and TTT have been found by Cryo-EM studies as well (Wang et al., 2017, eLife)). The mutagenesis in both AdeB and AcrB (the latter in the revised version) combined with susceptibility assays are comparing the roles of the homologous side chains in drug transport. They appear to be different.

During the revision stage, Morgan et al. published structures of AdeB trimers in different conformations with ethidium bound. It appears that in the absence of substrates AdeB is in the OOO or extrusion/extrusion/extrusion state, while in the presence of substrates, L (access), T (binding) and O (extrusion) states are apparent (and an additional "resting state"). We interpret from our data, obtained without substrate present, that the L* state is the initial state for substrate binding. Docking studies, included in the revised version of the manuscript, indicate that substrates can bind to the L* state. This would fit the proposed conformational cycle, since AdeB in the OOO state cannot bind drugs due to the lack of access (proximal) or deep binding (distal) pockets (and the lack of channels with a periplasmic entrance leading to them). Therefore the newly discovered L*OO state, might be the entrance state for drugs sequestered from the periplasm (or outer leaflet/inner membrane region).

Nevertheless, as the author mentioned, "The DBP of AdeB is highly lipophilic (13 non-polar residues, compared with 11 in AcrB and 4 in AcrD)...", the DBP is not necessarily conserved. This could explain why mutagenesis-based drug resistance behaved differently between AcrB and AdeB, and it is not surprising and makes the deduction from AcrB to AdeB not straightforward. For example, as the author claimed, "We suggest that in AdeB, R6G is shifted slightly in the DBP compared to its location in AcrB...", the binding mechanism for R6G is still inconclusive given the different performance in AcrB and AdeB.

Thank you for this comment. We conducted with a set of residues in AcrB additional mutagenesis studies using the same assay system (plate dilution). These new results indicate that R6G susceptibilities in AcrB is also very sensitive upon single side chain substitution. However, the results for TPP and especially levofloxacin/chloramphenicol are different from the AdeB mutagenesis at homologous positions. We therefore postulate that binding and/or transport for levofloxacin and chloramphenicol, possibly TPP (Table S6, revised version) is different between AcrB and AdeB. Moreover, the critical residue F610 in AcrB is, upon substitution with Ala, increasing susceptibilities for most AcrB substrates (Bohnert et al., 2008 and Table S6, revised version). This residue is T605 in AdeB. Substitution of T605 to Ala, shows diverse but only mild effects on susceptibility for all substrates (Table 1). Substitution of this residue to Phe (T605F) in AdeB, however, decreases the activity of AdeB toward polyaromatic substrates and increases for all other substrates (except doxorubicin). The effect on ETH resistance of the T605F is much less severe compared to R6G, whereas from the superimposed structures (Figure S14, R6G from AcrB co-structures, ETH from AdeB co-structures), there appears not a positional difference in the plane. The AdeB docking studies (Figure S7, revised version) indicate that R6G interacts with T605 indicating its adjacent position to this residue. The T605F substitution might therefore sterically affect the R6G binding. For ETH, shown to bind with two molecules in the deep binding pocket, the binding site most distance from T605 might not be affected, hence leading to a less severe resistance phenotype.

The corresponding residues could be located by sequence alignment, making the newly solved "OOO" and "LOO" structures of AdeB is not necessarily important compared to previous published AdeB structures.

The previous solved structure was in the OOO state (Su et al., 2019). From that structure, no implications could be made concerning drug binding or transport (the protomers have closed entrance channels and only the exit gate as seen for the O state in AcrB as well). The L*OO state (not LOO, since L* is not L) is much more revealing. It shows three entrance tunnels in the L* conformer towards a (closed) deep binding pocket. This is an original interesting observation and reveals that apparent asymmetry exists in AdeB and that the drugs can be sequestered via these channels. This is important since the OOO state cannot sequester drugs from the periplasm because of the lack of substrate binding sites. We hope the referee can agree that from primary amino acid sequences alone (even with homology modelling), one cannot derive to a substantiated insight of different conformations and binding mechanisms of these transporters. Structures are therefore absolutely necessary to gain insight into transport mechanisms of different RND transporters. The AdeB L*OO structure is therefore complementing the OOO structure and in line with the substrate bound LTO and TOO structures recently published during the revision stage of this manuscript by Morgan et al. (Edward Yu's laboratory).

It will help readers understand it by directly comparing the pore-forming three alpha-helices that connect the funnel in the middle. A statement of conformation is "in between "L" and "O" states" need to be more specific to demonstrate the comparison between this new structure with previous determined models both AdeB and AcrB.

Thank you for this comment. We agree. The definition "between the L and T states" was made by comparing the RMSD's of homologous RND transporters, in particular AcrB. The overall RMSD is lower between the AdeB L* with the AcrB T protomer than with the AcrB L protomer (Table S2, revised version). As shown in Fig. 2G, H of the revised version, the pore-forming helices (as part of the PN1 subdomain) are invariant between the L, L* and T states. The main difference between the L* and T state is the closed deep binding pocket. The orientation of the switch loop therefore allows binding of drugs to the L* state, which we think will trigger the T state as shown by the recently published TOO state (Morgan et al., 2021). We added supplementary figures (Supplementary Figure S5 and S6 in the revised version) with superimpositions between trimeric AdeB in the L*OO and OOO states and superimpositions between the L* state and the drug bound L and T states of the LTO AdeB trimer of the recently published AdeB structures (Morgan et al., 2021)

The mechanism could be better demonstrated if the author could make more use of the structure of AdeB, like 1. Biophysical experiments applied to AdeB would be needed to investigate/verify the binding mechanism proposed by the authors.

We agree, to continue testing the hypothesis, biophysical methods would be the way forward. Since the mechanism is dynamic and highly dependent on the other components of the pump (AdeA and AdeC), reconstitution of the tripartite system appears necessary. This would be subject for an entire PhD thesis.

2. Molecular dynamics or docking, based on the AdeB structure, could be conducted to verify the binding mode of at least one antibiotic.

Thank you for this suggestion. We collaborated with the University of Cagliari (Italy) on the docking of two substrates (R6G and ETH) to the L* conformer and have implemented these results into the manuscript (Table S4, Figure S7, revised manuscript). Molecular docking was further supplemented by molecular mechanics and free energy calculations.

Besides, there is some minor change required for revision:

1. Figure 2E is too mass in the background, needs to be replaced with a cleaner version.

We have modified Figure 2E to Figure 2E and 2F, according to your suggestion.

2. In Figure 2F, color make the reader not easy to recognize. The author needs to reconsider the color code used in the figures.

Thank you for this comment. We have made Figure 2F into Figure 2G and 2H for easier recognition.

3. In Figure 3 and Figure 4, the name of each substrate could be labeled on the figure to help reading.

Good suggestion, we included the names of the bound drugs in each subfigure.

Reviewer #2 (Remarks to the Author):

The paper entitled « Structural and functional analysis of the promiscuous AcrB and AdeB efflux pumps suggests different drug binding mechanisms » by Alina Ornik-Cha, Julia Wilhelm et al. reports structures of two homologous RND pumps, AcrB from the efflux pump AcrAB-TolC efflux pump in *E. coli* and AdeB from the AdeABC efflux pump of *A. baumannii*. The structures were respectively obtained by X-ray crystallography (in that case the protein was detergent-solubilized) and cryo-EM (after reconstitution of the protein into Salipro nanodiscs, providing a native-like lipid environment to the protein). In that study, authors also analyze the binding modes of AdeB and AcrB of nine substrates in light of three new crystal co-structures of AcrB periplasmic domain in complex with levofloxacin, doxycycline and fusidic acid.

The authors reach two main findings:

AdeB adopts two conformations in the conditions of the experiments: most of the particles are described in a resting state with all protomers in the O conformation (« OOO ») and a subpopulation of particles (30%) adopting a so-called L*OO conformation, in which one of the protomers is in a transition state between the L and T conformations, hence its denomination « L* ».

Different, previously unknown, binding modes are described for key residues in the binding of structurally diverse drug compounds thanks to in vivo activity assays.

The manuscript is well and clearly written and experiments are well designed. Results are convincing and novel. Results and discussion should also be of interest and beyond the efflux pump field, in particular, the discussion regarding binding properties and related conformational effects (conformational selection versus induced fit) is of general interest.

We like to thank the reviewer for the encouraging comments, and insightful remarks and review.

Hence we strongly encourage the editor to accept this paper for publication provided that the following comments and corrections are addressed.

lane 26: it is stated there that 10% of the protomers adopt the intermediate state. In the text (lane 152), it is written that 30% of the particles adopt the intermediate state. Of course, both statements are in perfect accordance but it would be clearer for the reader to adopt a single formulation.

Thank you for this comment. We changed the latter statement to “After classification of all protomers, we found that approximately 10% of the protomers adopt an intermediate state, i. e. 30% of particles adopted a trimeric arrangement of two O-protomers together with a previously uncharacterized conformation” (line 160-163).

lane 140: the authors report the structure of AdeB. Have they considered co-reconstituting AdeA ?

Yes, we considered co-reconstitution with AdeA/AdeC, to obtain insight of the AdeABC complex. This is an important further study, and to expedite the process, we would like to do this with the labs who have impressively showed the AcrAB-ToIC and MexAB-OprM tripartite structures.

lane 144: several tools are available for the stabilization and manipulation of membrane proteins for cryoEM studies. What was the specific advantage of the salipro methodology compared to SMALPs or amphipols ?

We started with detergent-solubilized AdeB, but due to a presentation at the faculty lecture series by Jens Frauenfeld, we immediately tried out the SaliPro reconstitution platform, which resulted in good quality particles.

lane 152: it is written in the Material and Methods section that two datasets were used to obtain the EM structure. The second one contained novobiocin. Did the presence of novobiocin affect the proportion of L*OO ?

Thank you for this comment. That was indeed our intention to see a change in the population by addition of substrate. The population distribution between L*OO and OOO trimers did, however, not change. We assume therefore that under the conditions tested, novobiocin did not bind to AdeB.

lane 154: It was recently shown that MexB, from the MexAB OprM of *Pseudomonas aeruginosa*, adopts a new asymmetric conformation where the exit pathway of the protomer involved in drug release is closed (dubbed « Closed conformation ») : ref: Glavier, M. et al. 2020 Nat Commun 11, 4948.

To what extent the L* state described here from AdeB compares to the C state described in MexB ?

Thank you for drawing the attention to these important publications. We included the references in the revised version. The L* state is suggested to be the initial state for drug binding, with three channels from the periplasmic side protruding into the porter domain of AdeB. The C state was suggested to be a state between the T and O conformations. There is an exit channel as seen in the O state, but an alpha-helical lid is blocking the end of the channel near the funnel domain, moreover, the PC2 subdomain deviates between the C state and O state. We address this point in the discussion, as the discovery of the C-state in MexB is an important observation for an intermediate state between the T and O state which were, together with the L state, the three main conformational states observed thus far. The recent AdeB trimer conformations (Morgan et al., 2021) also reported a “resting” state and structurally might resemble the C state or a state between O and L, but has to be analyzed in more detail. Intermediate states are supportive in the interpretation of the molecular mechanisms of these pumps. This has been in part discussed in Alav et al., 2021, recently, which we also added as a reference.

lane 157 to 163: RMSDs should be stated in Å.

Thank you for this correction. The unit has been included in the text and Table S2 of the revised version.

lane 181: at what cutoff are the EM maps displayed ?

The densities are displayed at cutoff levels in Chimera 0.0356 (OOO) and 0.0343 (L*OO). We have implemented this information in the legend of Figure 1.

manipulating the structures with Chimera, we find that:

_ chain a of 7B8P perfectly aligns with chain c of 7B8Q

_ chain c of 7B8P aligns with chain b of 7B8Q

_ chain b of 7B8P aligns with chain a of 7B8Q with significant RMSD.

Do the authors confirm that chain a of 7B8Q corresponds to the L* ? It would be interesting to add these information in the manuscript.

We confirm that chain a of 7B8Q is the L* conformation. We added this in the legend of Figure 2 of the revised manuscript.

lane 386: it would have been interesting to compare the effects of aminoglycosides. Did the authors consider doing so ?

Thank you for suggestion! We have tested aminoglycosides in MIC assays and found no clear difference in MIC in absence/presence of aminoglycosides. We also tried dye-efflux and dye-uptake assays and added aminoglycosides as putative competitive substrates to inhibit dye transport. We also conducted [³H]-gentamycin uptake studies. The results were, however, inconclusive and were not showing any indication of aminoglycoside binding, transport, or inhibition.

lane 493: it is described that the DBP of AdeB remains in the L* conformation in the absence of substrate although it is found open for AcrB in the same condition. Do the authors suggest that the L* conformation does not exist in AcrB ?

At the current stage, combining the structural and functional insights of both AcrB and AdeB, we indeed hypothesize that opening of the AdeB is subject to induced fit, i.e. presence of the (hydrophobic) substrate near the closed deep binding pocket induces the opening of the pocket. The latest structures of Morgan et al., 2021 appear to indicate that multiple drug binding (same drug, same protomer) is apparent in AdeB. In AcrB, we suggest a conformational selection, i. e. the L and T states are in equilibrium (but the L state is dominant) and substrate binding to the already open deep binding pocket stabilizes the T conformation (and removing it from the L/T equilibrium).

The structures of the AcrB LTO conformation without added drug appeared to be in line with this suggestion. Moreover, the AdeB structure paper published recently (Morgan et al., 2021), also shows AdeB in the OOO state in absence of bound drugs, whereas the drug-bound conformations are either in the TOO, RTO (R=resting state) and LTO state. Interestingly, the T states in any of these trimeric structures, always contain a drug molecule (ethidium) bound in the T protomer access pocket next to one (or two) ethidium molecules bound in the deep binding pocket. Moreover, the L state in Morgan et al. (2021) also contains an ethidium molecule bound in the access pocket. This observation might be in support of the L* state shown in this manuscript to be an initial state without drug bound from where, after initial drug binding as shown in the docking experiments, the protomer changes to the L state or even the T state (see Discussion in the revised manuscript).

lane 544 / lane 550 / lane 551: volumes are set as proportional to « grams ». We guess that is is per gram of wet cell pellet ...

Thank you. Yes, per gram wet weight cells. For the membranes: volume per gram wet weight membranes. We corrected this in the revised version.

lane 548 versus lane 569: pressures are defined in kPsi (lane 548) or bar (lane 569). please correct.

Done

lane 642: at what temperature is soaking performed ?

Soaking was performed at 18 degrees Celsius. We have indicated this now in the revised version.

lane 666: correct « Müller Hinton » for « Mueller Hinton ».

Corrected (also in the legend of the Supplementary figures/Tables).

lane 924, Figure S7B: do the authors know why AdeC displays two bands ?

We think that the second band might be due to not completely unfolded AdeC species. Similar effects have been seen for TolC.

lane 997: there seems to be a typo for construct #50.

The number refers to the primer number (also for #52, 56, 58, 60, 64, and 68)

The reason for this is that we list primer pairs (forward (_FW) and reverse (_RV)). For some of the mutant generation we used the same reverse primers (_RV). We will clarify this in the table by stating "Primer No. 30" etc.

This manuscript reports structure-functional analyses of AdeB from *Acinetobacter baumannii*. The structure of AdeB was solved using cryo-EM and twenty mutants in putative binding sites of AdeB were constructed. The manuscript also reports the ligand-bound structures of the periplasmic domain of AcrB from *E. coli*. These structures of AcrB are used as guides in selection of amino acid residues of AdeB for mutagenesis. The major conclusion is that binding sites of AdeB are different from those of AcrB. *A. baumannii* and its AdeB are important targets for drug discovery and the reported findings provide new insight into role of specific amino acid residues of AdeB in the ability of this transporters to reduce inhibitory concentrations of antibacterial agents. The manuscript is clear and concise. However, there are major deficiencies that need to be addressed.

We thank the reviewer for the in-depth review and helpful comments and suggestions

1. The impact of AdeB mutations is analyzed in the recombinant *E. coli* cells lacking *acrB*. The differences between *E. coli* and *A. baumannii* are substantial and the same mutants could have different phenotypes depending on the host. It is critical that at least some of the mutants are tested in *A. baumannii* background, see also below.

2. The properties of *A. baumannii* outer membrane are indeed quite different from those of *E. coli*. The authors emphasized the difference between porins but ignored the differences in the structures of lipid A in LPS. *A. baumannii* outer membrane is more permeable than that of *E. coli* to hydrophobic compounds, because *A. baumannii* utilizes fatty acids for growth. *E. coli* porins make its outer membrane more permeable than *A. baumannii* for hydrophilic molecules, but it is significantly less permeable than *A. baumannii* to hydrophobic compounds. Analyses of antibacterial activities in *E. coli* bias efflux pumps and they appear to be more efficient against hydrophobic molecules. Most of the authors interpretations about properties of substrates are biased by their permeation across the outer membrane. This is another reason why mutants should be analyzed in the context of the native host.

Thank you for this insightful comment. We assume that fatty acids are primarily crossing the OM via FadL and its homologs (van den Berg, *Curr Opin Struct Biol*, 2005), so the FadL pathway might not be a general pathway for hydrophobic compounds like antibiotics. For some hydrophobic compounds, it has been shown that OM permeation was higher for *A. baumannii* than for *E. coli*. We have added this observation in the Discussion with reference to the publication (Krishnamoorthy et al., 20017).

Concerning the background of the efflux pumps and the functional analysis: Whereas we agree that pumps might confer different phenotypes in the different bacterial setting, in this study our prime objective is to analyze the structure function differences between the AcrB and AdeB pumps.

To analyze the effects of substitutions on the activity and to compare the effects between AcrB and AdeB, the background has to be the same, i.e. not dependent on different OM permeabilities (or the activities from other pumps). We do not conclude on any phenotypical resistance effects for *Acinetobacter baumannii in vivo* (that would be indeed not valid for the reasons mentioned), but only on the difference in mechanism of drug transport in AcrB and AdeB.

The *E. coli* setting is the next best thing compared to an *in vitro* reconstitution, which is at the moment is not routinely feasible for testing different substrates. Even if the *E. coli* outer membrane is less permeable towards hydrophobic compounds and more towards hydrophilic substrates, this affects cells harbouring AcrAB-TolC and AdeABC in the same way. Any difference in sensitivity can be directly related to the activity of the pumps.

I like to refer to a study our lab has done with the MFS-transporter CraA (Foong et al., 2019, *J Antimicrob Chemother.*). This transporter was initially characterized in *A. baumannii* (Roca et al., 2009, *AAC*), where the deletion of the *craA* gene only shows (considerable) effect on the MICs of chloramphenicol (hence the name chloramphenicol resistance *Acinetobacter*).

We confirmed this *craA* phenotype for *A. baumannii*. However, when *craA* was expressed in *E. coli*, it showed a multiple resistance profile, not only conferring resistance towards chloramphenicol, but also against benzalkonium, chlorhexidine and dequalinium amongst other drugs, which is not apparent in the *A. baumannii* background. This observation was in line with the homology of CraA with the multidrug efflux transporter MdfA from *E. coli*. Clearly, the background of *A. baumannii* obscured the true drug preferences of CraA, which became visible once the gene was transferred in an *E. coli* Δ mdfA Δ emrE background.

3. Authors devote significant attention to aminoglycosides both in Introduction and in Discussion. Indeed, resistance to aminoglycosides is one of the signature features of AdeB. However earlier studies (see Leus et al., 2018) showed that AdeB overproduction is required but not sufficient for aminoglycoside resistance of *A. baumannii*. Authors do not report the results with aminoglycosides. But these are important. The speculative statements in Discussion about possible alternative roles of AdeB in aminoglycoside resistance need to be substantiated by data.

We thank the reviewer for the reference for Leus et al., 2018, we missed to cite in the manuscript. We included this reference in the revised version. In Leus et al. it was also stated that chloramphenicol and aminoglycoside susceptibility was not changed while AdeABC was overproduced. We have indeed tried to address aminoglycosides both in MIC assays (*E. coli* expressing AdeABC) as well as dye-efflux and uptake assays, by adding aminoglycosides as putative competitive substrates for dye transport. We also conducted [³H]-gentamycin uptake studies but results for all three assays were inconclusive.

On the speculative statements in the Discussion section: Since the role of AdeABC in aminoglycoside resistance is controversial, we agree that the role of AdeABC in aminoglycoside resistance has to be resolved first. In this light, we remove the section in the discussion, and formulate a more nuanced statement in the introduction on the role of AdeABC in aminoglycoside resistance, including the reference to Leus et al., 2018.

4. The structures of AcrB are not integral to the presented results and their inclusion into this study is somewhat awkward. The sites are different between AcrB and AdeB. Ligand-bound structures of AdeB would be more revealing. Inclusion of AcrB mutants, see below, could make the study more cohesive.

Agree, co-structures would have been extremely helpful, but we were not able to get data with substrate bound to AdeB (we tried with novobiocin). Meanwhile ethidium-bound AdeB structures have been published (Morgan et al., 2021) and are implemented as comparison in the manuscript.

We also included plate dilution assays on cells harbouring AcrB variants and their susceptibilities for comparison in the revised version.

5. Comparison of transporters' activities in *E. coli* is very unreliable, because of differences in the expression levels between native and recombinant AdeABC. Most of the AcrB residues in binding sites were already analyzed in previous studies. Inclusion of at least some of AcrB variants with mutations in analogous positions could make the comparison in *E. coli* host more reliable. MIC measurements are needed to make the data more compatible with previous studies.

The comparison between AdeB and AcrB in *E. coli* appears reproducible, also the expression appears for most of the AdeB variants equal in the plate dilution assay (Figure S10-12 in the revised version).

Even if the expression levels between WT AcrB and WT AdeB are not exactly similar, it does not explain the lower growth rates observed for AcrB for R6G, ETH and especially TPP (Table 1, first row "AcrB" which displays "AcrB minus AdeB" data).

In fact, as an example, AdeB variants F136A and Y327A (Figure S11 and Table 1) show increased levels of resistance compared to wt AdeB against chloramphenicol and levofloxacin (Figure S11A), matching the level inferred by the AcrB wildtype. Although wildtype AdeB is expressed at equal levels as these AdeB mutants (Figure S11B), wildtype AdeB confers much less resistance. The opposite is observed for substrates R6G and TPP (Table 1, Figure S11A). The Western blot samples are all from prepared membranes of those cells to ensure we detect membrane-inserted AdeB. This example as well as results for other variants (e. g. position E89 or E151) indicate that the differences seen between the activity of AdeB wildtype and AcrB is not due to lack or difference of expression. In our opinion, this is a robust assay system to compare wildtype AdeB and variants.

We agree that comparing the data with MIC data from earlier publications with the plate dilution assay (PDAs) is not ideal. We have therefore conducted susceptibility assays for cells harbouring AcrB variants (Figure S10 in the revised version).

In our experience, the plate dilution assays (PDAs) are much more sensitive toward changes in activity for different mutants compared to MIC assays. Often growth on substrates/drugs exhibit MIC changes of one dilution step (WT vs. inactive control), which do not allow for substitution variant analysis. For diagnostics, we think MIC assays are the gold standard at the moment, but to analyze mutants which infer slight changes in activity, we experienced reproducible and consistent data with clearer differences in susceptibility when conducting PDAs.

As an example, the results on the AcrB variants revealed that cells harbouring F610A are highly susceptible towards all substrates tested (in accordance with the MIC observations in Bohnert et al., 2008). For R6G, all AcrB substitutions caused a considerable effect. One report (Yao et al., 2013, PMID: 23627437) listed the effect of the Y327A variant in a different *E. coli* strain and with a different expression system on R6G susceptibility (MIC). The MIC value in that study was clearly lower, but still clear above the MIC value for the negative control. For other drugs, like chloramphenicol and TPP, moderate effect on growth was observed (one-dilution step difference). This exemplifies the difficulties by using MIC assays for efflux pump variants and PDA assays offer more nuanced insights as shown in Table 1 and in Table S6.

For the AcrB variants, we analyzed the plates (3 biological replicates, Fig S10 in the revised version) by three persons (one PhD student who conducted the assay, one undergraduate student not involved in the study, and one supervisor) to count the number of spots on each plate. From the 84 experiments, 74 times the persons counted the same number of spots and in the other 10 cases, 1 person counted one spot less or more. We average these numbers, which results in the tables (Table S1, Table S5). The results from the original submission were done in a similar way, except that the counting was done by the first and second author, and the supervisor (all counting was done independently, i. e. without information of the spot counting from the other persons)

REVIEWERS' COMMENTS

Reviewer #1 (Remarks to the Author):

I agree with the author the AdeB L*OO conformation makes more impact on the importance of the paper. The author has added data/experiments requested. I strongly suggest the editor accept publishment.

Reviewer #2 (Remarks to the Author):

The authors have addressed all our comments and the manuscript is now much improved. The results are convincing and provide an important contribution to the field. From our point of view, the revised manuscript can be published in Nature Communications.

Reviewer #3 (Remarks to the Author):

The authors responded constructively to the previous criticism and the revised manuscript is notably improved with new insights. However, the manuscript is difficult to read due to the overuse of abbreviations and very descriptive presentation of mutagenesis data. The authors are encouraged to reduce the number of abbreviations, to streamline the logic of mutagenesis results and to make the manuscript more accessible to readers outside of the immediate field. Minor issues noticed:

1. p. 4, l. 96 - references are needed.
2. p. 6 - consider adding paragraphs.

REVIEWERS' COMMENTS

Reviewer #1 (Remarks to the Author):

I agree with the author the AdeB L*OO conformation makes more impact on the importance of the paper. The author has added data/experiments requested. I strongly suggest the editor accept publication.

Reviewer #2 (Remarks to the Author):

The authors have addressed all our comments and the manuscript is now much improved. The results are convincing and provide an important contribution to the field. From our point of view, the revised manuscript can be published in Nature Communications.

Reviewer #3 (Remarks to the Author):

The authors responded constructively to the previous criticism and the revised manuscript is notably improved with new insights. However, the manuscript is difficult to read due to the overuse of abbreviations and very descriptive presentation of mutagenesis data. The authors are encouraged to reduce the number of abbreviations, to streamline the logic of mutagenesis results and to make the manuscript more accessible to readers outside of the immediate field.

We have used abbreviations for the drug substrates, like LFX (levofloxacin), DOX (doxorubicin) etc., we have changed this now in the revision, using the full names at every occasion. This will reduce the number of abbreviations substantially, leaving abbreviations like i. e. the access pocket (AP) and deep binding pocket (DBP). The subdomains PN1, PN2, PC1, PC2 are the given names for the subdomains (Murakami et al., 2002) and are indicated e .g. in Figure 2. Due to the number of substitutions and the interaction of drugs with the different side chains, description may appear verbose (very descriptive). However, we also provided paragraphs describing e. g. the shared binding site of ethidium, rhodamine 6G and tetraphenylphosphonium or the substitutions causing an improved activity of the pump compared to the wildtype pump. We also think that the data are best analyzed when the corresponding pdb file of the respective structure is used in a graphical program (e. g. PyMOL) and the side chains described are highlighted in the structure. A polyspecific pump is characterized by its many binding modes depending on its substrate bound and it involves many binding partners (side chains). The challenge is indeed to keep track of the various phenotypes in the different variants, which we have put an effort in this work to describe and combine similar phenotype patterns in relation to the side chain substitutions.

Minor issues noticed:

1. p. 4, l. 96 - references are needed.

The reference given is Frauenfeld et al., 2016 (nr. 28)

2. p. 6 - consider adding paragraphs.

We have added paragraphs in this section